# Dynamic decorrelation as a unifying principle for explaining a broad range of brightness phenomena

**Alejandro Lerer**[1], **Hans Supèr**[1,2,3,4], **Matthias S. Keil**[1,2] *

**1** Departament de Cognició, Desenvolupament i Psicologia de l'Educació, Faculty of Psychology, University of Barcelona, Barcelona, Spain, **2** Institut de Neurociències, Universitat de Barcelona, Barcelona, Spain, **3** Institut de Recerca Pediàtrica Hospital Sant Joan de Déu, Barcelona, Spain, **4** Catalan Institute for Advanced Studies (ICREA), Barcelona, Spain

☯ These authors contributed equally to this work.
* matskeil@ub.edu

**Data Availability Statement:** All relevant data are within the manuscript and its Supporting Information files.

**Funding:** AL was supported by the Ministry of Economy and Competitiveness (Spain) PSI2010-

## Abstract

The visual system is highly sensitive to spatial context for encoding luminance patterns. Context sensitivity inspired the proposal of many neural mechanisms for explaining the perception of luminance (brightness). Here we propose a novel computational model for estimating the brightness of many visual illusions. We hypothesize that many aspects of brightness can be explained by a dynamic filtering process that reduces the redundancy in edge representations on the one hand, while non-redundant activity is enhanced on the other. The dynamic filter is learned for each input image and implements context sensitivity. Dynamic filtering is applied to the responses of (model) complex cells in order to build a gain control map. The gain control map then acts on simple cell responses before they are used to create a brightness map via activity propagation. Our approach is successful in predicting many challenging visual illusions, including contrast effects, assimilation, and reverse contrast with the same set of model parameters.

## Author summary

We hardly notice that what we see is often different from the physical world "outside" of the brain. This means that the visual experience that the brain actively constructs may be different from the actual physical properties of objects in the world. In this work, we propose a hypothesis about how the visual system of the brain may construct a representation for achromatic images. Since this process is not unambiguous, sometimes we notice "errors" in our perception, which cause visual illusions. The challenge for theorists, therefore, is to propose computational principles that recreate a large number of visual illusions and to explain why they occur. Notably, our proposed mechanism explains a broader set of visual illusions than any previously published proposal. We achieved this by trying to suppress predictable information. For example, if an image contains repetitive structures, then these structures are predictable and will be suppressed. In this way, non-predictable structures stand out. Corresponding mechanisms act as early as in the retina (which

18139-P, HS was supported by Ministry of Science and Innovation (Spain) PGC2018-096074-B-100, MSK was supported by Spanish Government Grant PGC2018-099506-B-I00. The funders had no role in study design, data collection and analysis, decision to publish, or preparation of the manuscript.

**Competing interests:** The authors have declared that no competing interests exist.

enhances luminance changes but suppresses uniform regions of luminance), and our computational model suggests that such mechanisms also might be used at subsequent stages in the visual system, where representations of perceived luminance (=brightness) are created.

## Introduction

Visual perception is relative rather than absolute; the visual system (VS) computes the perceptual attributes of a visual target not only based on its physical properties, but also by considering information from the surrounding region of the target (context). For example, it is possible to induce different kinds of effects by context modification, such that the brightness of a target is contrasted (increasing brightness differences) or assimilated (decreasing brightness differences) with respect to its adjacent surround (e.g. [1]). Variants of these effects give rise to a myriad of visual illusions, which are of great utility for building hypothesis about computational mechanisms or perceptual rules for brightness perception.

At first sight it seems that contrast effects, such as simultaneous brightness contrast (SBC; Fig 1A), can be explained by lateral inhibition between a target (center) and its context (surround). However, activity related to brightness contrast does possibly not occur before V1, albeit the receptive fields of retinal ganglion cells are consistent with lateral inhibition [2].

Unlike brightness contrast effects, brightness assimilation (e.g. Fig 1C and 1D) pulls a target's brightness towards to that of its immediate context, and therefore cannot be explained by mechanisms based on plain lateral inhibition. In fact, the neural mechanisms involved in generating brightness (perceived luminance) and lightness (perceived surface reflectance), respectively, appear to be more intricate (e.g., [3–7]). Intrinsic image approaches are (computational) models which aim at deriving further images based on the characteristics of the *depicted visual scene* of an input image [8]. With respect to lightness computations, corresponding algorithms estimate a (sometimes chromatic) reflectance image and a shading image from an input image (e.g. [9–11]). Further approaches with respect to intrinsic images in a wider sense address surface qualities such as transparency (e.g., [12]) or gloss (e.g., [13]).

The majority of models for computing brightness estimates decompose an input image typically with filters of different orientations and multiple spatial frequencies (image-based decompositions, [8]). In this way, a set of images (=filter response maps) is derived, from which the input can be recovered. The recovered image is considered as brightness prediction. For instance, feature-based approaches classify filter responses into lines and edges (e.g. [14–18]), and build a brightness prediction based on recognized features.

It is particularly challenging for computational proposals to simultaneously explain contrast and assimilation effects with identical model parameters. Blakeslee and McCourt, [19, 20], proposed a highly successful image decomposition with oriented difference of Gaussians (ODOG) filters. In the ODOG-model, a brightness prediction is generated by two steps. First, a weighted sum of filter responses across spatial frequencies is computed for each orientation (orientation channels). Second, each orientation channel is divided by its root mean square level before they are summed to yield the final brightness map. Although the ODOG model predicts SBC and many assimilation displays, in our re-implementation it fails at the Benary Cross (Fig 1D) and Chevreul's Illusion.

Several variants of the multiscale-decomposition approach have been proposed. Dakin and Bex, [21], showed that isotropic filters are sufficient to reproduce the Craik-O'Brien Cornsweet effect and White's effect. Otazu and co-workers, [22], used an invertible wavelet

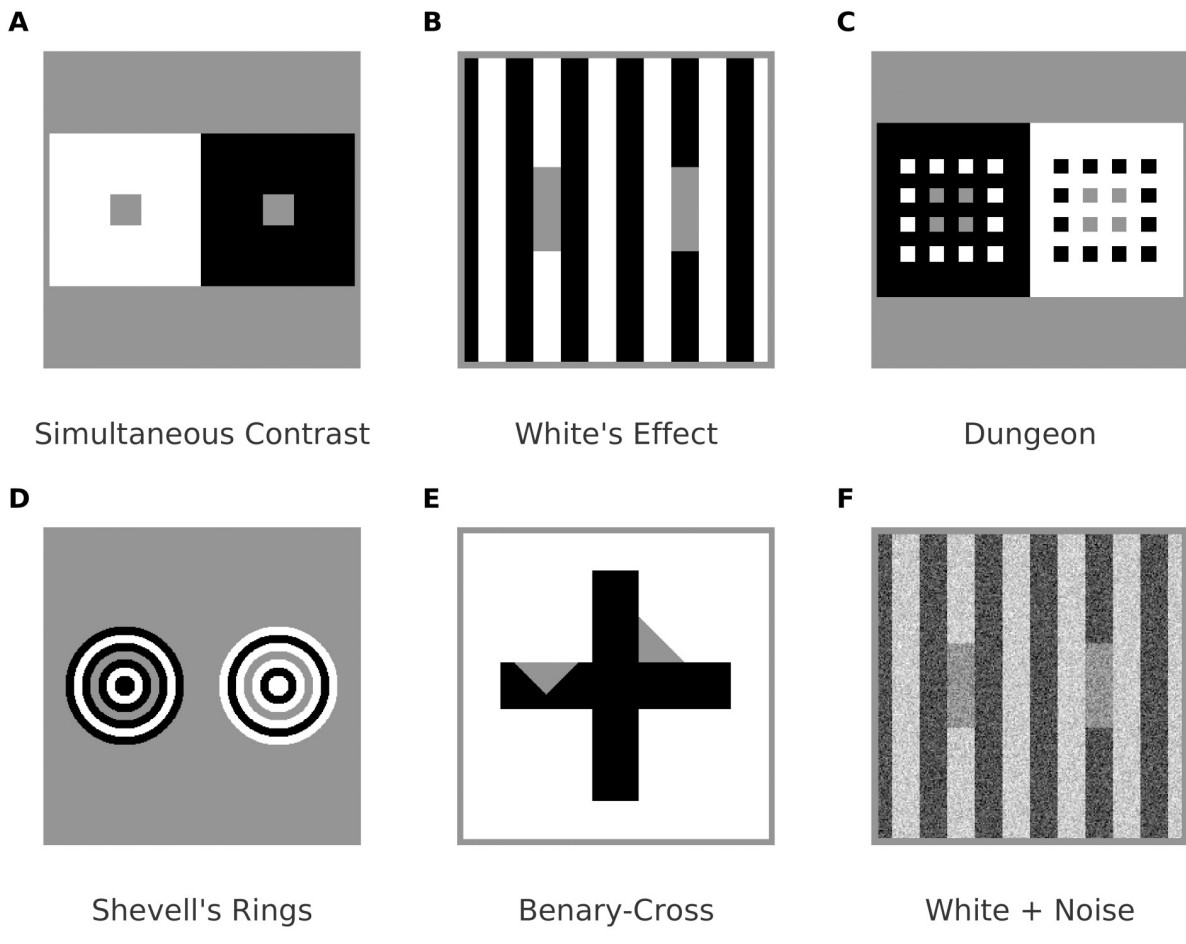

**Fig 1.** (A) Simultaneous brightness contrast: The two gray patches with identical luminance increase their brightness difference with their respective backgrounds. (B) White's effect is consistent with contrast at the horizontal contours of the bars, and with assimilation along the vertical contours. (C, D) Examples of brightness assimilation: The gray structures with identical luminance decrease their brightness difference with their respective background. (E) Sensitivity to context: The gray triangles have identical luminance and selectively contrast with the cross (left) or the background (right). (F) White's effect is still intact in the presence of noise.

transformation with wavelet coefficients being weighted according to the contrast-sensitivity function [23]. The weights are further adjusted dependent on local surround contrast. However, models based on adjusting the spatial frequency distribution of filter responses appear to fall short of predicting brightness when the input images are contaminated by (band-pass filtered) noise [24], (Fig 1F).

Computationally, the spatial operation of certain types of retinal ganglion cells can be dumbed down to taking the second spatial derivative of the visual input. In this way, sharp contrasts (edges or boundaries) are enhanced. Edge integration models [1, 25–27] try to invert spatial differentiation (=contrast extraction) by thresholding (suppress shallow gradients due to illumination effects), and integration (across edges in order to estimate lightness). This means that edge integration models—unlike multi-scale approaches—propose explicit mechanisms for suppressing illumination effects, and for estimating a reflectance image. Filling-in (FI) models [28, 29] can be considered as a neural implementation of edge integration, where boundary activity is propagated laterally in order to generate perceptible surface properties.

Whether or not activity propagation plays a role in surface perception is subject of a yet ongoing debate [6, 30, 31]; also, where it might occur in the visual system (e.g. [32, 33]). It is often argued that with Grating Induction, FI-mechanisms would incorrectly predict a homogeneous test-field, because activity would "average out". This argument, however, ignores that boundary webs could form across the test field [34], and/or darkness and brightness channels could interact [35, 36]. Furthermore, Grating Induction seems to be perceived instantaneously, what seems incompatible with activity propagation at the neuronal level [6, 37]. However, this argument ignores that the activity to be propagated can be initialized at a coarser scale than the boundaries that contain it. Furthermore, FI may proceed simultaneously in layers with different resolutions [38]. This would render the perceptual effect indistinguishable from multi-scale filtering. Finally, it has been argued the addition of noise would disrupt activity propagation [6, 21]. This argument ignores, however, that texture-like features such as noise (even symmetric simple cells with small receptive fields) and surface boundaries (odd symmetric features) may be represented by different layers [36]. Furthermore, even with a single representation, noise would likely not form closed domains in order to contain activity (e.g. Fig 7 in [35, 39]), or would not fill in at all if the initial activity is computed at a coarser scale.

Typical FI models distinguish two types of contours [40]. The first type represent barriers for activity propagation, and defines the boundary contour system (BCS). It represents the complete 3-D boundary structure of a visual scene, including boundaries from texture and depth. The second type is the feature contour system (FCS) and represents surface properties to be filled-in, such as brightness, lightness, color and depth. FCS processing was suggested to occur in cytochrome-oxidase staining regions in V1 (blobs) and thin stripes in V2. Interaction between FCS and BCS was hypothesized to occur in V4 [41]. The hypothetical FCS/BCS dichotomy is compatible with several experimental findings [42–46].

Whereas the prediction of brightness contrast usually is straightforward with (most) filling-in architectures, assimilation effects remain challenging. On the basis of a one-dimensional luminance profile, Grossberg and Todorovik (in [40]) explained how two (non-)adjacent (luminance) regions could influence each other. If boundaries are sufficiently near in the BCS, then their activity is reduced. Therefore, activity propagation in the FCS may not completely be blocked, causing FCS activity to fill into (non-)adjacent surfaces. In this way, one surface may not just be influenced by the brightness of its immediate surrounding region, but even from further away.

Domijan, [47], extended these ideas to two dimensions. He computed luminance-modulated FCS activity with an unbalanced center-surround kernel, similar to [48], but see also [35] for a different way of luminance encoding. BCS activity is computed by first deriving a local boundary map, where the loss of activity at junctions and corners was corrected. Based on the local boundary map, a global boundary map was computed. In the latter, contours which are parallel or co-linear to another contour were enhanced. Finally, local boundary activity was divided, at each position, by global boundary activity. The division is approximately one at those positions where no contour enhancement took place in the global boundary map (otherwise it is smaller). The final BCS output keeps only those activities that are relatively close to one—boundaries with smaller activity which are parallel to high contrast edges are eliminated. This causes FCS activity to freely diffuse across the eliminated boundaries. In this way, Domijan was able to predict 2-D assimilation displays with a FI-architecture.

Ross and Pessoa, [49], modified FCS activity before filling-in by using an occlusion-sensitive copy of the BCS. The modification of the original boundaries is based on T-junctions: Boundaries along the stem of the "T" are suppressed, while the others are enhanced. The modified boundary map ("context boundaries") is subsequently used for suppressing contrast measurements in the FCS. The original boundaries act as diffusion barriers in the FCS. Although

the model successfully predicts White's effect and the Benary Cross (Fig 1B and 1E), psychophysical evidence suggests that White's effect seems not to be affected significantly if the T-junctions are suppressed [50, 51] (cf. Fig 1D), nor seem to be other illusions [52]. Furthermore, it is not readily clear whether junction rules do represent reliable cues in complex natural scenes: The utility of junctions rules has only been illustrated with relatively simple artificial displays [49, 53].

Barkan [54] used center-surround receptive fields at four resolution levels for edge (or contrast) extraction. At each resolution level, filter response amplitudes ("local contrast") were gain-controlled with a low-pass filtered version of themselves ("remote contrast"). A brightness map was estimated from the gain-controlled contrast map with fixpoint iteration of a Laplacian [55], which implements the filling-in process. The model was successful in simulating assimilation and reverse assimilation effects (mostly centered on challenging variants of White's effect), but failed in predicting Simultaneous Brightness Contrast (SBC).

A completely different approach for explaining visual illusions is based on a statistical analysis of real-world images. This approach suggests that the perception of brightness [56, 57] or lightness [57–59] is related to knowledge about the statistical relationships between visual patterns across space. In particular, [56] proposed that the brightness of a visual target embedded in some context depends on the expected luminance according to a probability distribution function. The probability distribution function integrates all contexts in that what which the target was seen previously. The perception of the target then depends on its expected luminance given its current context: It is perceived as brighter if the expected luminance is lower, and it is perceived as darker otherwise. This approach is successful in predicting contrast and assimilation for several visual illusions, and suggests a statistical relationship between luminance patterns and brightness perception. Unfortunately, no attempt has been made in order to unveil any information processing strategy from the statistical analysis (but see [59]).

As with some of the models reviewed above, our approach also emphasizes the importance of boundaries in brightness perception: We propose to reduce redundancy in the boundary maps. Such encoding strategies usually reduce the overall activity of a representation and thus the expenditure of metabolic energy, [60, 61], and are also known as efficient coding [62], predictive coding [63], whitening [64] or response equalization [65].

With respect to mid- or higher-level processing, White [66] suggested that a pattern-specific inhibition mechanism acts in the visual cortex, which inhibits regularly arranged patterns of a visual stimulus. Our model is related to White's idea: We adjust a boundary map, such that redundant activity is suppressed, while non-redundant activity is enhanced. Since neurons that encode redundant patterns tend to be over-represented, the overall boundary activity is reduced after the adjustment (response equalization). Response equalization is carried out by a dynamic filter.

Fig 2 shows an overview of our model. In the first step an input image is encoded by two sets of Gabor filters, which mimic the spatial response properties of simple cells in V1 [67]. The responses of the high-resolution filters define the Contrast-only channel (similar to the BCS), while responses of the more coarse-grained filters define the Contrast-Luminance channel (similar to FCS). Here the term 'channel' refers to a feature of our model. We do not imply two pathways for encoding luminance and contrast in the visual cortex. From the Contrast-only channel, we compute boundary activity via local energy [16, 17]. Local energy is insensitive to the phase information, and thus resembles complex cell responses. From the local energy map, a decorrelation kernel is learned, and then applied to it, in order to reduce redundancy (=dynamic filtering). The redundancy-reduced energy map then functions as a gain control map for both contrast channels. As a consequence, contrast activity is modified. Subsequently, an iterative procedure is used to recover a brightness map from the two contrast

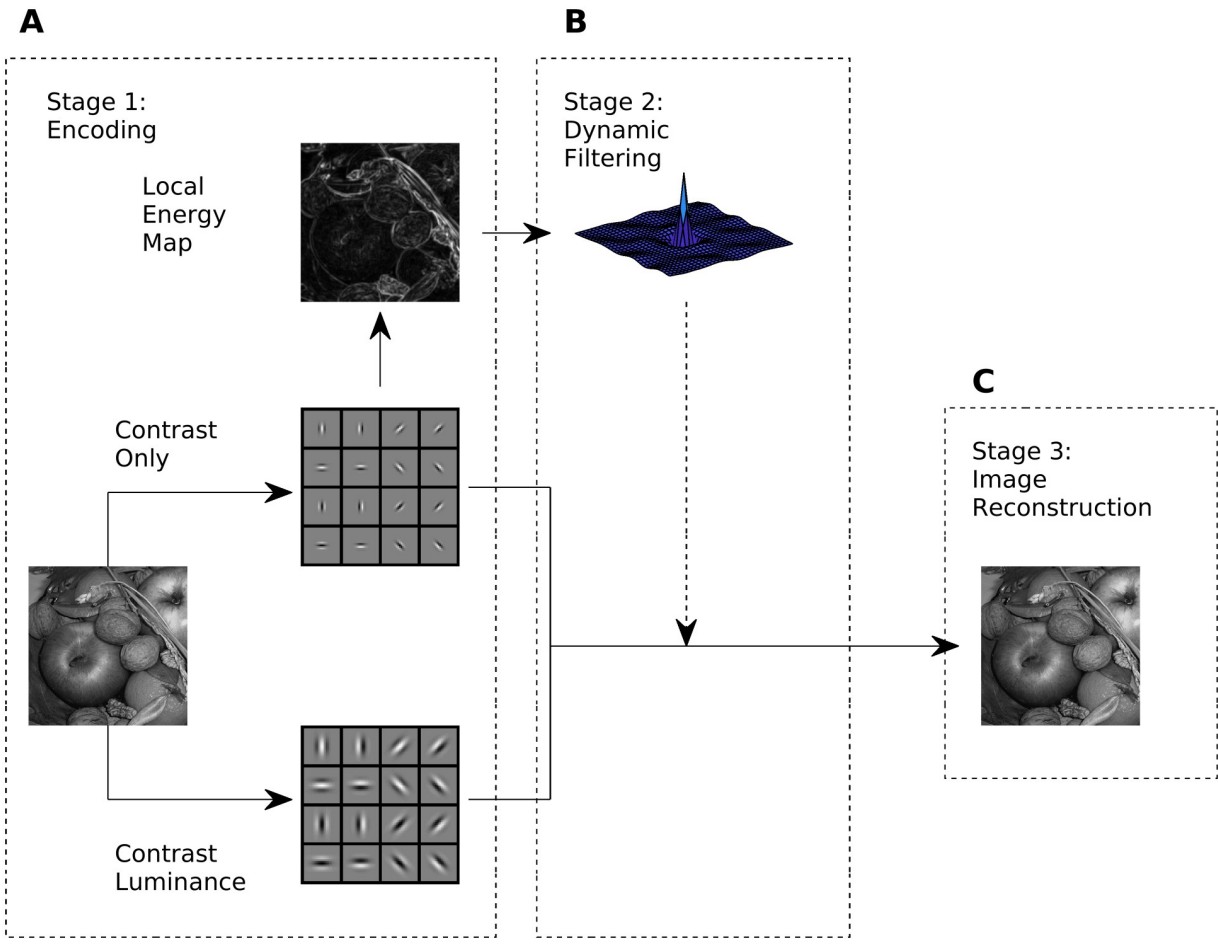

**Fig 2. Model overview.** Each of the three stages is mathematically specified in the Methods Section. (A) Stage 1: The Contrast-only channel and Contrast-Luminance channel are instantiated by filtering an input image with a corresponding set of Gabor filters with high spatial resolution (0.25 cycles per pixel) and coarse resolution (0.125 cycles per pixels), respectively. The local energy map is computed from the Contrast-only channel. (B) Stage 2: The kernel of the dynamic filter is estimated from the local energy map. Dynamic filtering equalizes the amplitude spectrum of the energy map, reducing redundancy. The decorrelated energy map serves as gain control for both contrast channels. (C) Stage 3: The output of the model is a brightness map that is obtained by solving an inverse problem, that is recovering the image from both contrast channels. Note that the two contrast channels do not interact with each other before Stage 3.

channels. Our iterative procedure resembles a filling-in process. The plausibility of our model is underlined by predicting many challenging visual illusion, including some that were never predicted by any other computational model so far.

## Materials and methods

Fig 2 depicts the three stages of our model. Stage 1 encodes the input image into a Contrast-only and a Contrast-Luminance channel by two respective set of Gabor filters. Stage 2 the dynamic filtering process. It consists of equalizing response amplitudes of the local energy map and then gain-controlling both channels. Finally, Stage 3 refers to the filling-in process for estimating a brightness map. The brightness map represents the output of our model. The three stages are detailed in the following subsections.

## Stage 1: Encoding

**Contrast-only and contrast-luminance channel.**   We use Gabor filters for encoding Contrast-only and Contrast-Luminance information. In the primary visual cortex, simple cells respond to oriented light-dark bars across a certain spatial frequency range [68] and their receptive fields can be modeled by Gabor filters [67, 69, 70]. Consistent with the properties of Gabor filters, it seems that many simple cells in V1 encode contrast information. Under certain circumstances though, neurons in V1 may respond to surface brightness as well, even without (sharp luminance-)contrasts in their receptive fields [71, 72]. For example, [73] found such neurons in V1 which have large receptive fields, broad orientation tuning, and a preference for low spatial frequencies. These neurons respond to both contrast and luminance.

In our model, the set of Gabor filters for the Contrast-only channel had a spatial frequency of 0.25 cycles/pixel and balanced ON-OFF sub-regions (i.e., the sum across the kernel is zero) In this way they did not respond to homogeneous regions of the input (DC-free). For the Contrast-Luminance channel we used Gabor filters with a lower spatial frequency (0.125 cycles/pixel), and unbalanced ON-OFF sub-regions (i.e., the sum across the kernel is positive) such that they respond to both luminance and contrast (non-zero DC part). Strictly speaking, these filters are actually no longer pure bandpass filters. But since the sum across the kernel is very small, the bandpass properties dominate the response. The use of bigger kernels which are selective for even lower spatial frequencies would not alter significantly our results. However, the computational cost would increase. For this reason, we chose the kernels of the Contrast-Luminance channel with half the spatial frequency of the Contrast-only channel. Fig 2A illustrates the two filter sets. Parameter values and a mathematical description of unbalancing the ON/OFF subregions is provided in section A in S1 Text.

The responses of the Contrast-only and the Contrast-Luminance channel were computed by convolving (symbol "∗") a luminance image with the corresponding set of Gabor filters. That is, if $g$ represents a Gabor kernel (either from the Contrast-only or the Contrast-Luminance channel), then $R_g$ represents its activity in response to the input image as:

$$R_g(x, y) = g * Im(x, y) \tag{1}$$

The arguments (x,y) denote 2D spatial coordinates. The contrast channels remain separated until the filling-in process. Fig 3A shows examples of filter responses. Notice that contrast responses dominate the response map of the Contrast-Luminance channel in Fig 3A. Fig 3C shows filter responses of the two channels to a luminance step (contrast) and a uniform region: The DC-response of the Contrast-Luminance filter is small compared with its contrast response.

**Local energy map.**   The local energy map resembles the properties of complex cells in the primary visual cortex [74]. Complex cell responses are similar to those of simple cells in terms of orientation and spatial frequency preference, but tend to be non-linear and shift-invariant with respect to contrast phase [74, 75]. Local energy can be calculated from a pair of Gabor filters in quadrature phase by summing their squared responses and then taking the square root [16, 17, 76, 77]. A quadrature pair are two Gabor filters with 90 degree phase difference, but with identical preference in orientation and spatial frequency. Finally, the local energy map E was computed through averaging the activity of our model complex cells across all orientations. Fig 2A shows an example local energy map. It essentially corresponds to the contours of the input image. Mathematical details are given in section B in S1 Text.

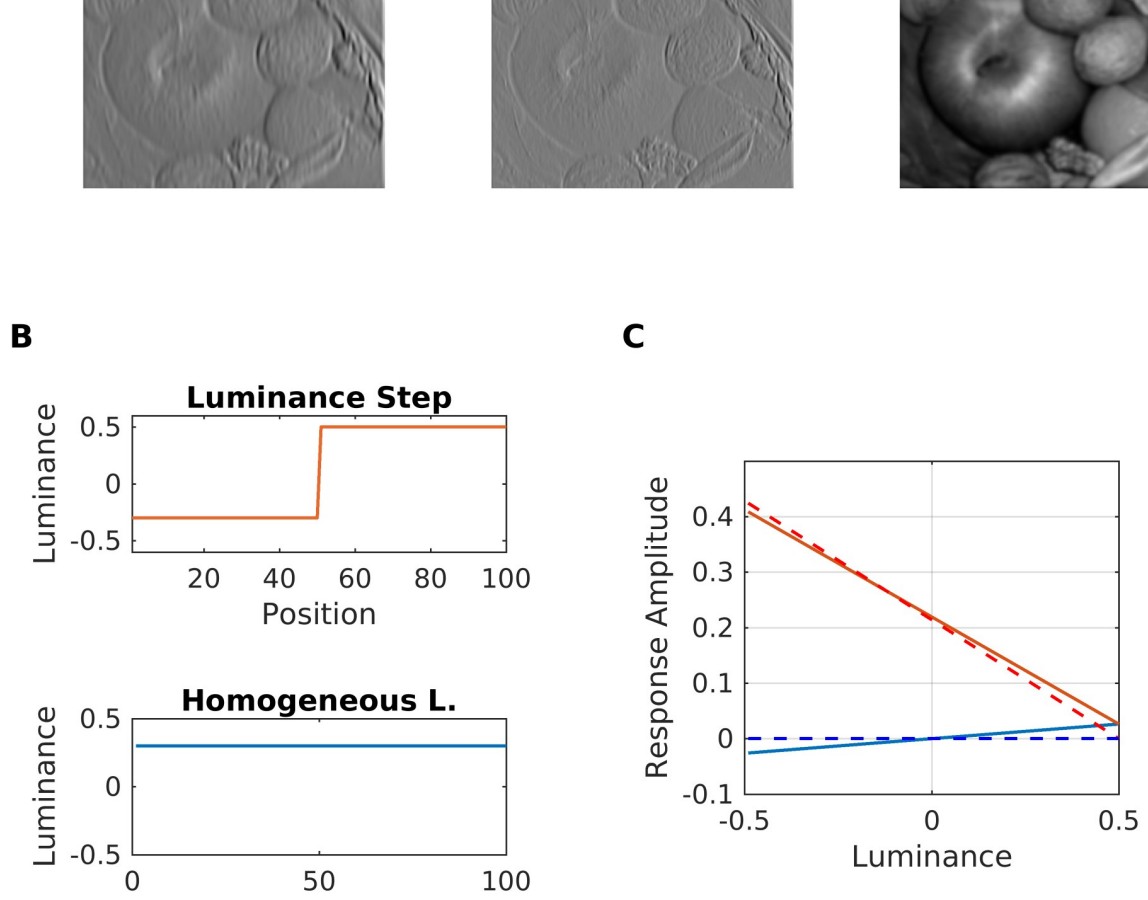

**Fig 3. Sensitivity to contrast and luminance.** (A) (left) Fruits filtered by a Contrast-Luminance filter g (Eq 1). (Middle) Fruits filtered by a Contrast-only filter g with the same orientation (Eq 1). (Right) Fruits filtered by a low-pass filter. (B) Example luminance step (values -0.3 to 0.5) and an example for a homogeneous region with luminance 0.3. (C) Responses (maximum) of Contrast-Luminance filters (solid lines) and Contrast-only filters (dashed lines) to the homogeneous region (red color) and the luminance step (blue color). For the luminance step, the lower luminance was increased from -0.5 to 0.5. Negative filter responses result from the use of negative input values.

## Stage 2: Decorrelation

In this section we describe stage 2 of our model (Fig 2B). The first subsection describes how the dynamic filter is computed with zero-phase component analysis (ZCA: [78]). ZCA is a decorrelation method that whitens the covariances of the original data while preserving their original direction [79]. With the dynamic filter we equalize the amplitude spectrum of the energy map. In fact, it produces very similar results to the "Whitening-by-Diffusion" method proposed in [65]. In the second and third subsections we detail the computation of the gain control map and how it interacts with the whitened energy map, respectively.

**Dynamic filter.** The purpose of the dynamic filter is to equalize the amplitude spectrum of the energy map. It is computed with zero-phase whitening (ZCA), a technique which has been used for learning the receptive fields of retinal ganglion cells [78]. ZCA resembles principal component analysis (PCA), and signal decorrelation can be achieved with both of the latter. However, the components are constrained to be symmetrical with ZCA. This "symmetry constraint" guarantees that the principal components are localized in the spatial domain [78], and therefore can be used as filter kernels. We nevertheless introduced a couple of modifications to the original ZCA (see section C in S1 Text). As a result of the modifications, we obtained a spatial filter that adapts to the spatial structure of the local energy map of an image. It is called "dynamical" because a different filter is learned from each image. After filtering, the amplitude spectrum of the energy map is more uniform (see Fig 4). By the Wiener–Khinchin theorem, a more uniform power or amplitude spectrum implies that the original signal is more decorrelated [80, 81]. For the decorrelated energy map, this means that spatial patterns with low redundancy tend to be intensified, while patterns with high redundancy tend to be attenuated. This is illustrated with Fig 4, where after filtering, horizontal edges are intensified in the energy map as compared to vertical ones.

**Gain control map.** The gain control map G is computed in two steps. First, the dynamic filter F is used as a convolution kernel for the energy map E:

$$\tilde{E}(x,y) = \begin{cases} F * (E(x,y) - \text{mean}(E)) & if \ E(x,y) \geq \omega \\ 0 & if \ E(x,y) < \omega \end{cases} \tag{2}$$

Here, the symbol "∗" indicates convolution. We set the threshold to 10 percent of the maximum activity as $\omega = 0.1\max(E)$. We observed that without thresholding, artifacts and noise tend to accumulate in the brightness prediction. In addition, thresholding increases sharpness. The exact percentage value is not critical. Our results would not change significantly when using, for example, 0.15, 0.2, or 0.3 times the maximum. In the second step we normalized the activity of the gain control map with a sigmoid function $S(x, a, b) = 1/(1 + e^{-ax-b})$ as

$$G(x,y) = \begin{cases} 2\left[S\left(\frac{\tilde{E}(x,y)}{max|\tilde{E}|}, a, b\right)\right] - 1 & if \ \tilde{E}(x,y) \neq 0 \\ 0 & if \tilde{E}(x,y) = 0 \end{cases} \tag{3}$$

The parameters were fixed as $a = 5$ and $b = \min(3\text{mean}(\tilde{E}), 0.3)$. Notice that the gain control map $G$ is normalized to $[-1, 1]$. The sigmoid's slope a controls the intensity and smoothness of $G$. For bigger values of a the soft-thresholding turns into hard thresholding. Even with hard thresholding, the results will be not significantly affected. For small values ($a < 1$), the Gain control map was practically useless due to the small values of G. The inflection point b (if $a \gg 1$ it would be the threshold) is, however, important for estimating brightness, as it determines when the values of G change from negative to positive: If $G(x, y)>0$, then the brightness contrast at $(x, y)$ is increased, while contrast will be reduced (=assimilation) if $G$ is negative. We found $b$ by manual optimization.

**Feedback interaction and channel gain control.** The Contrast-only and the Contrast-Luminance channel were subjected to gain control using the decorrelated energy map as

$$R_g^*(x,y) = R_g(x,y)\left[\frac{\tau + \tau G(x,y)}{\tau + |R_g(x,y)|G(x,y)}\right] \tag{4}$$

Here, $G$ is the gain control map, $\tau$ is a control parameter, $R_g$ represents the activity of a Gabor filter $g$ of the corresponding filter set (i.e., Contrast-only or Contrast-Luminance), (x,y)

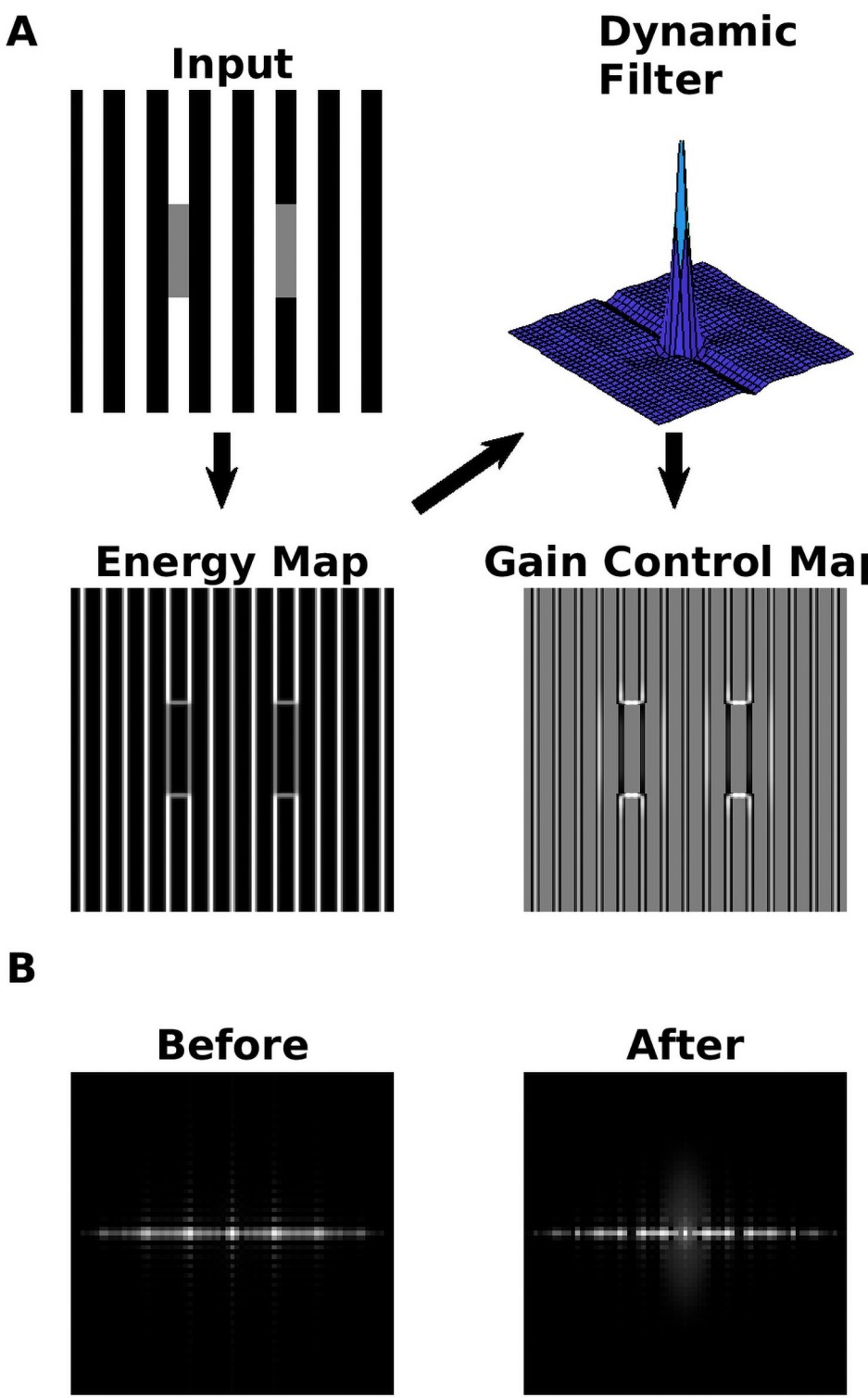

**Fig 4. Action of the dynamic filter.** (A) The arrows indicate the steps in order to obtain the gain control map: (i) A local energy map is computed; (ii) a dynamical filter is constructed with a customized zero-phase whitening procedure (ZCA, see section C in S1 Text); (iii) a gain control map is obtained by filtering the energy map with the dynamical filter (see subsection Gain Control Map). (B) The power spectrum (= square of amplitude spectrum) before and after of applying the dynamic filter on the local energy map.

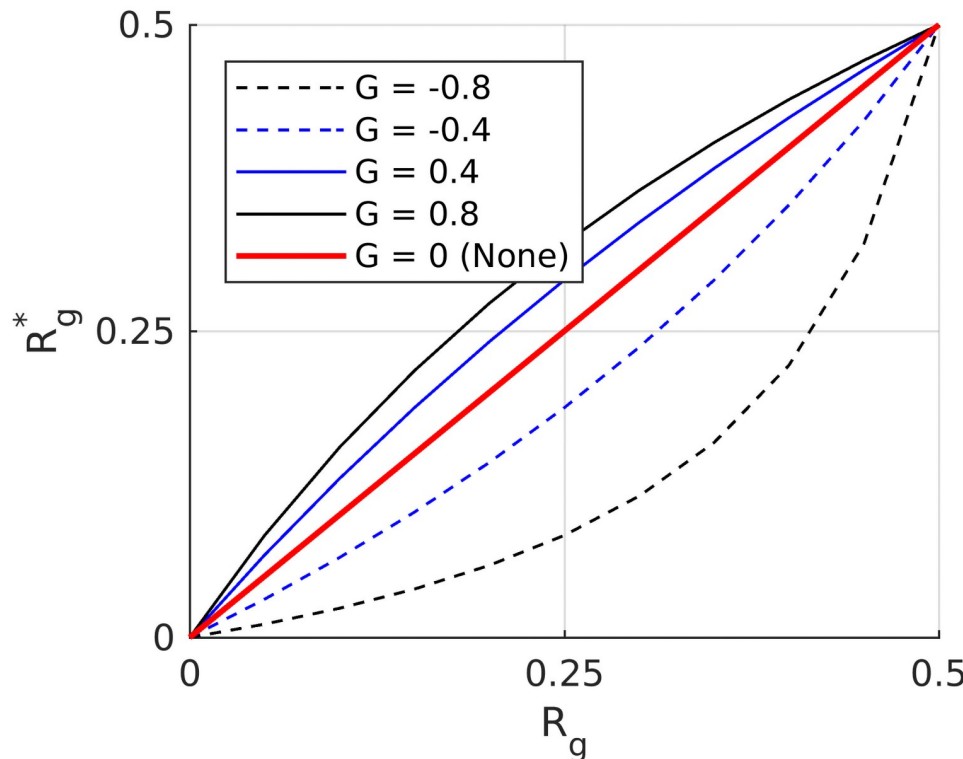

**Fig 5. Filter responses before and after applying Eq 4.** Each curve represents a different scalar value for the gain control G (notice that in Eq 4, G is two-dimensional). The values of G corresponding to each curve are indicated in the figure legend.

denote pixel coordinates, and $\tau$ is a control parameter. The control parameter $\tau$ acts as an upper bound for the maximum activity that any filter can reach when it encodes a luminance grating (min=-0,5, max = 0.5) that matches the spatial frequency of the filter.

Fig 5 illustrates the behavior of Eq 4 for different values of the gain control map $G$, where $G$ was set to the (scalar) values indicated in the figure legend. Gabor filter responses are amplified for $G > 0$. For $G < 0$, they are attenuated. The gain control map $G$ is furthermore modulated by the denominator of Eq 4. The modulation is weak or absent when Gabor filter responses $R_g$ are equal to $\tau$ or 0, (Fig 5), while it is strong between the latter values. The modulation is crucial for explaining Chevreul's illusion and Mach Bands, but it is not relevant for all other brightness displays. Finally, after applying Eq 4 to each channel, the results were lowpass filtered with a Gaussian kernel (standard deviation 1 pixel) in order to reduce possible artifacts.

## Stage 3: Brightness estimation as a filling in process

The brightness map $\hat{z}$ is the output of our model. It is estimated by minimizing an objective function $E(z)$, which optimizes the trade-off between the reconstruction error (first term in the sum of Eq 5) using the gain-controlled contrast channels $R_g^*$ and a smoothness constraint (second term):

$$\hat{z} = \mathrm{argmin}_z E(z) := \mathrm{argmin}_z \sum_g \| R_g^* - g * z \|^2 + \mu \| \nabla^2 z \|^2 \tag{5}$$

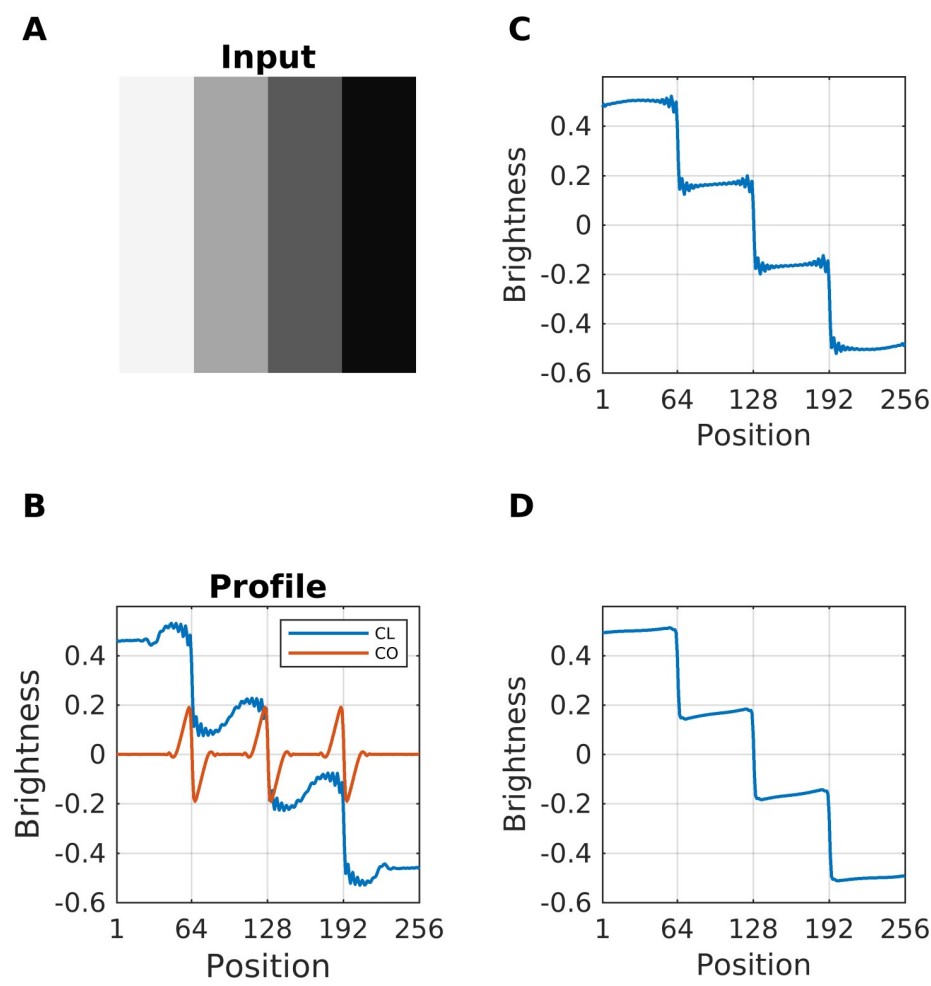

**Fig 6. Contribution of each channel to brightness estimation.** (A) A luminance staircase (giving rise to Chevreul's Illusion) served as input. (B) The resulting brightness profiles are estimated at 10 iterations. The blue curve (legend label "CL") uses only the Contrast-Luminance channel in Eq 5. The red curve ("CO") uses only the Contrast-Only channel without responses to luminance. (C) Resulting brightness profile at the stop criterion for the Contrast-Luminance channel. Aliasing artifacts appear due to undersampling. (D) Brightness estimation with both channels. The Contrast-only channel eliminates undersampling.

Notice that the sum involves the Gabor filters $g$ of both contrast channels (i.e., Contrast-only and Contrast-Luminance). The regularization parameter controls the smoothness constraint and is set to 0.01. The Laplacian is denoted by $\nabla^2$. The smoothness term and the Contrast-only channel serve to reduce artifacts produced at discontinuities (Fig 6). The equation is solved iteratively with the conjugate gradient method. The method starts with an image $z_0$ of random values. The weights of $z_k$ are updated such that the value of the cost function $E(z_k)$ decreases with each iteration step. This gradient descent continues until a maximum of 100 iterations is reached, or until an error criterion is satisfied (see section D in S1 Text for more details). The filter responses $R_g$ determine how the weights $z_k$ are updated. Because the $R_g$ are higher at the edges, activity is iteratively propagated from the edges. Direction and orientation of activity spreading are determined by the odd-symmetrical Gabor filters that are used for encoding the image. Thus, image reconstruction (i.e., brightness estimation) proceeds according to a filling-in process.

It is instructive to highlight the role of both channels for estimating brightness. The bulk of activity propagation depends on the Contrast-Luminance channel. Since it has bigger filter kernels than the Contrast-only channel, it propagates activity across greater distances per unit time. The time to convergence is further reduced by the residual luminance responses of the kernels in the Contrast-Luminance channel. Fig 6 compares the reconstruction of a luminance staircase at 10 iterations based on a single channel. Although both channels are used in Eq 5, the major contribution to brightness estimation comes from the Contrast-Luminance channel, while the Contrast-only channel serves to encode edge-information. As a consequence, the Contrast-only channel could be excluded from the reconstruction without significantly affecting brightness predictions. Nevertheless, the participation of the high-frequency filters (Contrast-only channel) eliminates undersampling (or sub-sampling), which would cause an accumulation of oscillatory artifacts (due to aliasing) close to the edges (see Fig 6C versus Fig 6D).

### Classification of model predictions according to three scenarios

The effect of the Gain Control Map on estimated brightness (=model output) is as follows. If a Gabor response amplitude after gain control has increased (decreased), it would produce a major (minor) contrast in the reconstructed image (=estimated brightness). This means that brightness estimates as generated by our model depend critically on Eq 4 (see also Fig 5). The purpose of this section is to evaluate the influence of dynamic filtering and Eq 4 on predicted brightness. To this end, we identified three prominent scenarios for explaining corresponding classes of brightness illusions (Fig 7).

The luminance pattern giving rise to Scenario 1 differs only in its spatial layout (Fig 7A), as all structures have the same intensity value. In this case, the patterns with high spatial correlations are attenuated by the dynamic filter, while patterns with lower spatial correlation are somewhat increased such that a brightness contrast effect in predicted for the central disk. Scenario 2 is defined by luminance patterns with similar spatial structure but different intensity range (see Fig 7B). Here the effect is limited by the size of the dynamic filter. We observed that the dynamic filter will not only reduce the spatial correlations, but it will also act as a contrast filter, if the redundant activity is in a sufficiently small spatial region. As a result, redundant activity with higher (lower) intensity than the other patterns would be increased (decreased). If this increment (or decrement) is sufficiently big, it will produce a major (minor) brightness contrast effect.

In Scenario 3, the major contribution to predicted brightness is caused by the Contrast-Luminance channel and modulation (denominator of Eq 4, see Fig 5). Edges in the Contrast-Luminance channel might be enhanced via the Gain Control Map. The enhanced edges eventually produce a boost in (estimated) brightness contrast. The degree of boosting depends on the ratio between the activity (after boosting) and the control parameter (upper bound) in Eq 4. An example of this effect can be observed by comparing both input images and their profiles (at the edges) in Fig 8. It is essential to highlight that Scenario 3 serves just to explain Chevreul's illusion and Mach Bands (see results section), but is not relevant for all other brightness displays.

### Results

All of the following simulation results are based on the same set of model parameters. This means that model parameters were never changed. This section presents simulation results (i.e. brightness predictions) from our model. The first subsection focuses on contrast effect: Simultaneous Brightness Contrast, Benary Cross and Reverse Contrast. The second subsection is

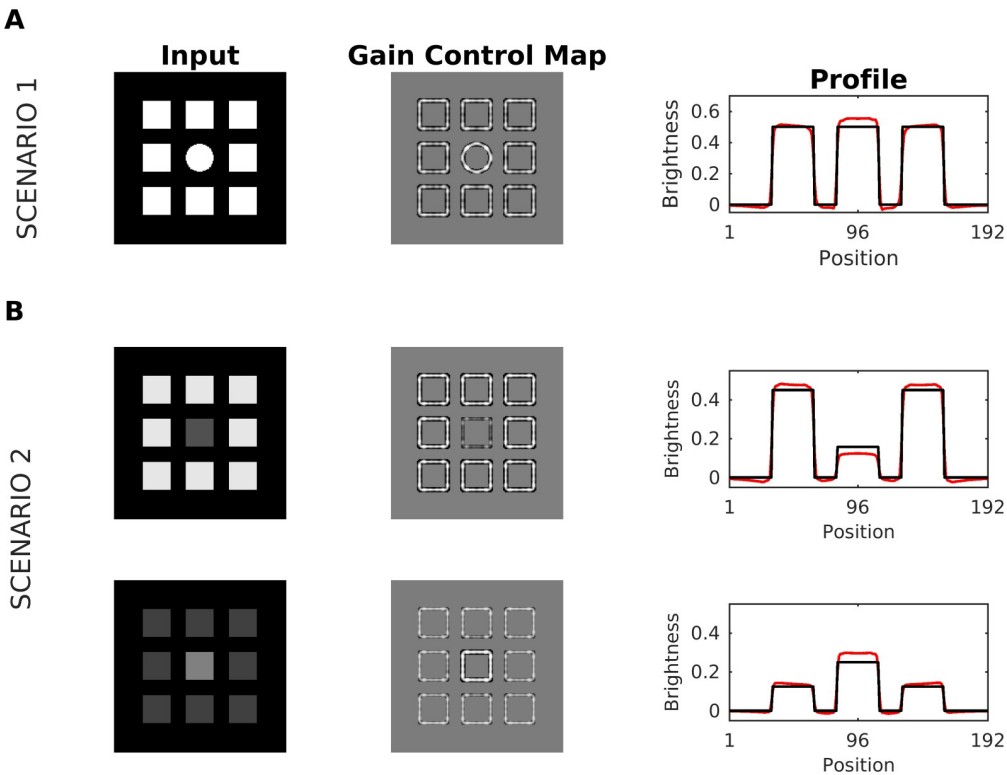

**Fig 7. Scenario 1 and scenario 2.** (A) Scenario 1: A disk embedded in a redundant pattern of eight squares served as input (first column). The middle column depicts the corresponding gain control map G, and the right column the profiles of input (black line) and brightness estimation (red line). The brightness of the center disk ts enhanced with respect to luminance, meaning that a brightness contrast effect is predicted merely based on redundancy (but not on grounds of luminance—note that all features have the same luminance). (B) Scenario 2. (Top) The input consists of a series of nine squares arranged in a spatially redundant pattern, where the middle square has a different luminance. The profile plot suggests an overall increase in brightness contrast: Brightness of the middle square is further reduced, while the brightness of the surrounding squares is enhanced. (Bottom) While the brightness contrast also increased in the display with the bright middle square, this increase in contrast is caused nearly exclusively by the middle square.

dedicated to assimilation effects: White's effect, Todorović's Illusion, Dungeon illusion, Checkerboard illusion, and Shevell's Rings. The third subsection shows our brightness predictions for the Craik-O'Brien-Cornsweet effect, Hermann/Hering grid, Chevreul's illusion (including the luminance pyramid), Mach Bands, and Grating induction. Finally, the last subsection shows how our model deals with Real-World Images and Noise. It is essential to highlight that all input images were normalized such that pixel intensity ranged between −0.5 and 0.5.

## Brightness contrast effects

**Simultaneous brightness contrast (SBC).** The SBC display consists of two gray patches with identical luminance which are embedded in a dark and bright background, respectively. The patch on the bright background is perceived as darker than the patch on the dark background (Fig 7A). SBC can be attributed to low-level processing. For example, retinal ganglion cells may enhance patch contrast by lateral inhibition. However, other studies suggest that SBC may involve higher-level processing as well [82]: The apparent brightness of the patches can be modulated by the region surrounding the patches (=spatial context). In fact, psychophysical studies report that the contrast effect is perceived more intense for smaller patches [83–87]. Fig 9 shows the estimated brightness for SBC. In our model, the effect conforms to

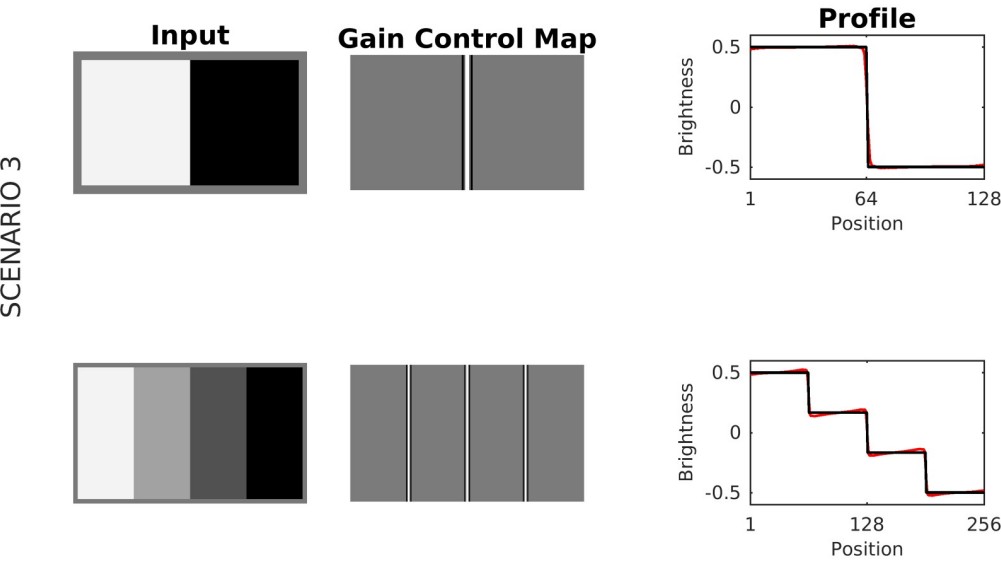

**Fig 8. Scenario 3.** A luminance step and a luminance staircase, respectively, served as input images. Activity in response to the luminance step is close to the control parameter of Eq 4 ($\tau = 0.5$), producing barely changes in the corresponding brightness estimation at the edges. In contrast, for the luminance staircase, the activity at the edges is relatively far from the control parameter, inducing a boost (an increment of brightness contrast) in the corresponding brightness estimation.

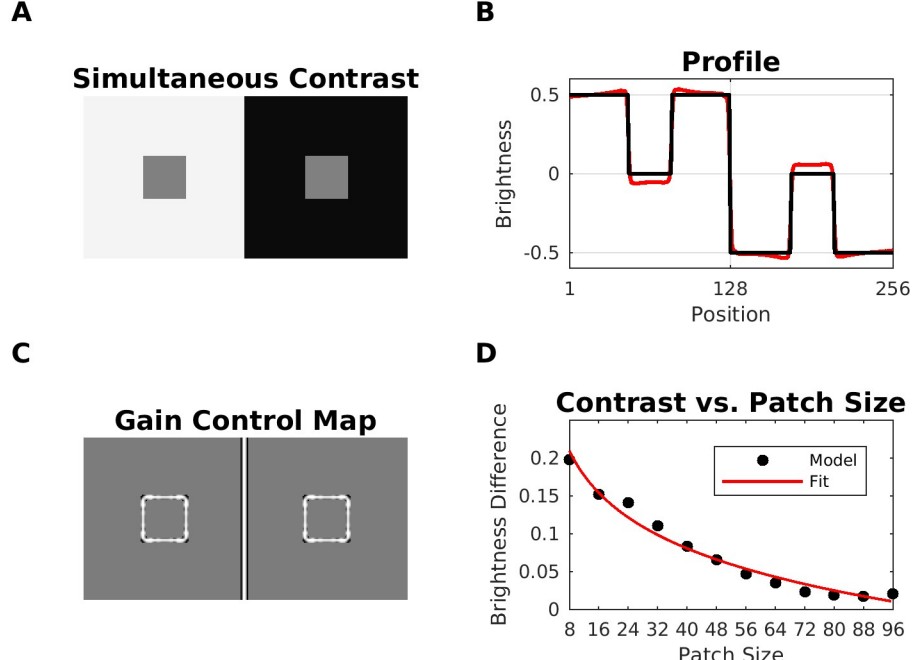

**Fig 9. Model prediction for simultaneous brightness contrast (SBC).** (A) Simultaneous brightness contrast display (model input). (B) The corresponding Gain Control Map. (C) Profile plot of the estimated brightness map (red line) and the input (black line). (D) Mean absolute brightness difference between the left and the right patch as predicted by our model (filled circles). The continuous (red) lines show the fit of $y = a + b\log(x)$ to the model data. The fit was carried out by linear regression with fitting parameters: intercept $a = -0.3851$, slope $b = -0.0823$, $R^2 = 0.9831$, and $RMSE = 0.0086$.

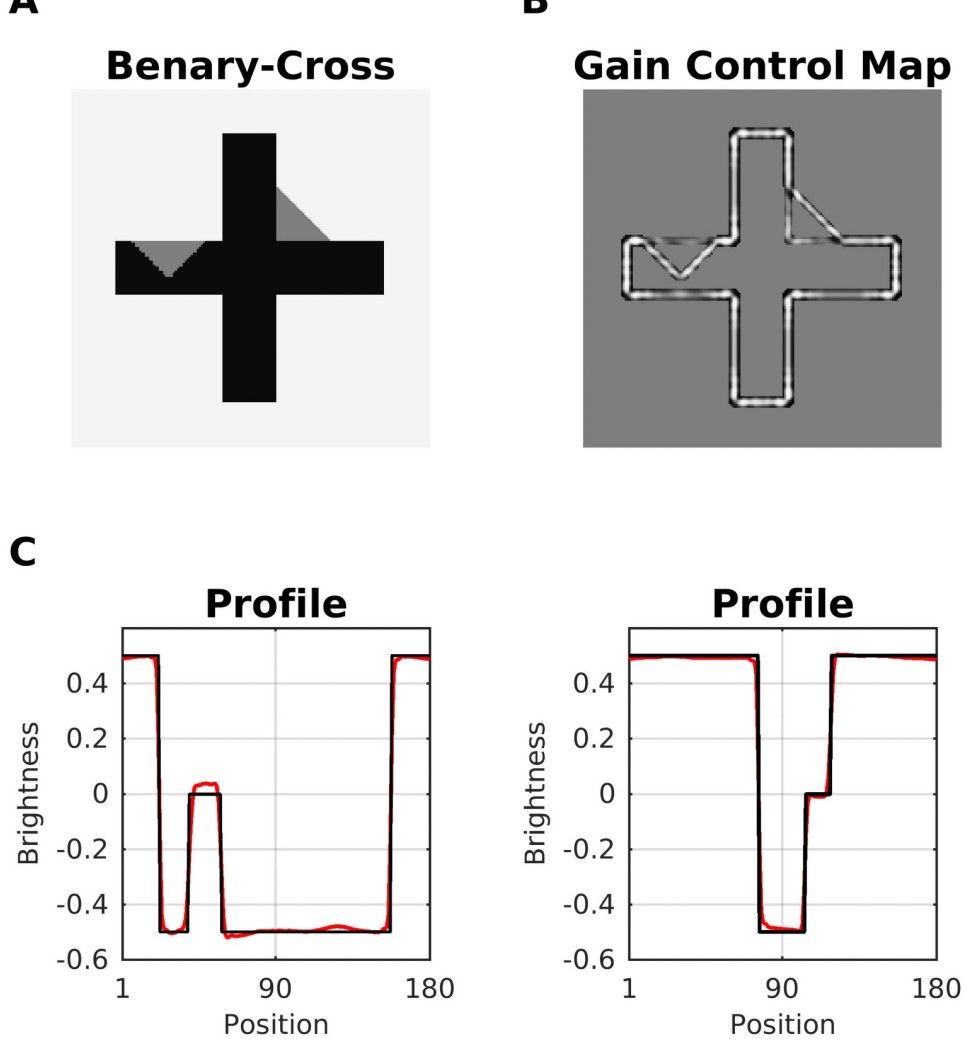

**Fig 10. Prediction for Benary Cross illusion.** (A) Benary Cross (input). Both triangles have the same intensity, but the triangle embedded in the cross is perceived as brighter. (B) In the gain control map, redundant edges (=aligned with the cross) of the triangle are weakened. (C) Profile plots of predicted brightness (red line) versus luminance (black line). The left profile plot shows the left triangle.

Scenario 1. In SBC, the patterns with low spatial correlation are the patch edges with equal intensity, what causes an enhancement of their contrast after gain control (Gain Control Map: Fig 9C). This translates to an increased contrast in predicted brightness (profile plot in Fig 9B). We also studied the relation between patch size and their predicted brightness. In agreement with previous studies, Fig 9D shows a logarithmic relationship between patch size and our brightness estimation [88].

**Benary cross.** The Benary Cross [89] is composed of a black cross and two gray triangles with the same luminance (Fig 10). The triangle embedded in the cross is perceived as brighter than the other. Notice that both triangles have identical contrast edges—one white to gray and two black to gray. This effect cannot be explained by lateral inhibition and is usually attributed to "belongingness theory", where the region in which the triangle appears to belong to induces a contrast effect [89]. Noise masking experiments support the idea that the effect is caused by

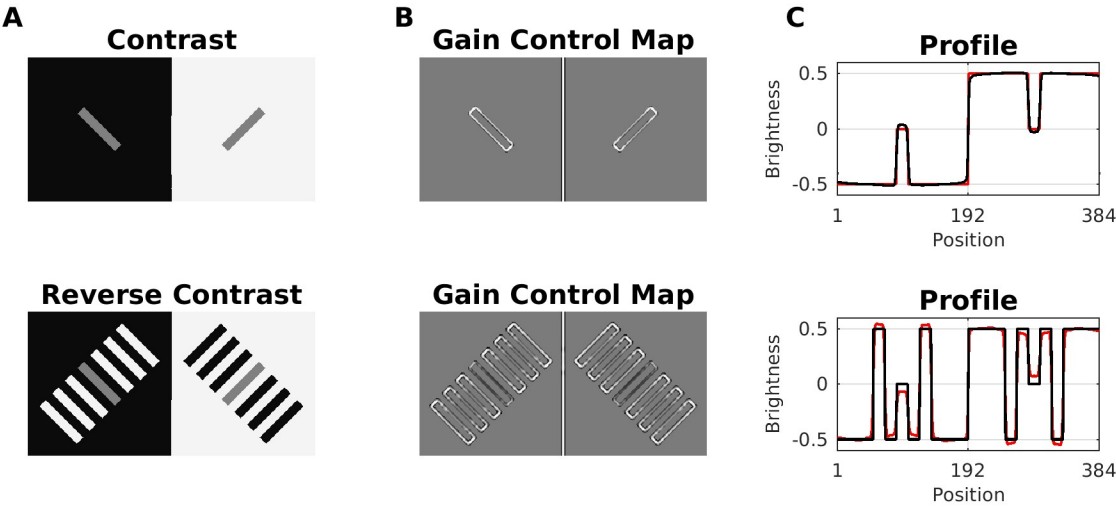

**Fig 11. Model prediction for reverse contrast effect.** (A) Simultaneous Brightness Contrast (SBC, top) and Reverse Contrast (bottom) which is constructed by adding flanking bars to the SBC configuration. All gray patches have the same luminance. Reverse contrast can be explained either by assimilation with the in-between bars that have the same intensity as the background, or as contrast with the flanking bars that have the opposite luminance to the background. (B) Gain Control Maps obtained by dynamic filtering. Notice the suppression of parallel edges corresponding to flanking patches with the same intensity. (C) Profile plots of predicted brightness (red line) versus luminance (black line).

low-level mechanisms [90]. Our model predicted the brightness difference in the triangles according to two scenarios. According to Scenario 1 (but also 2, considering the intensity differences), the redundant patterns correspond to those edges of the triangles which are aligned with the cross. These are attenuated, while the non-redundant edges are enhanced. The Gain Control Map suggests (Fig 10) a rather balanced effect, which is confirmed by the model's predicted brightness map. Notice, however, that the length of the non-suppressed edges is bigger for the left triangle.

**Reverse contrast.** Gilchrist & Annan (in [91]) suggested that simultaneous brightness contrast (SBC) can be reversed (e.g. by overcoming lateral inhibition) by adding more structures to the original SBC display. This is Reverse Contrast (Fig 11). The purported mechanism acts on grounds of perceptual grouping of these structures.

Our model predicted the reverse contrast effect according to Scenario 1 and 2, respectively. In case of SBC, dynamic filtering increments the activity of the non-redundant edges that outline the two patches (Gain Control Map: Fig 11B). In the case of reverse contrast, the redundant activity depends on both edge orientation (Scenario 1) and contrast polarity (Scenario 2). Accordingly, all parallel edges of the (flanking) patches with equal intensity are weakened by the dynamic filter. However, the central patch has a different intensity than the flanking patches, and its edges are enhanced. In order to better understand how our model predicted reverse contrast, we probed it with further configurations (see Fig 11). We observed that the change in brightness of the gray patches increases as a function of the number of flanking bars (Fig 12A). On the other hand, if the flanking bars were misaligned to various degrees (disrupting the good continuation principle of perceptual organization), the effect was considerably reduced (Fig 12B). Both results stand in agreement with psychophysical experiments [92]. However, in the latter study the authors examined displays with even more configurations that our model cannot predict (results not shown).

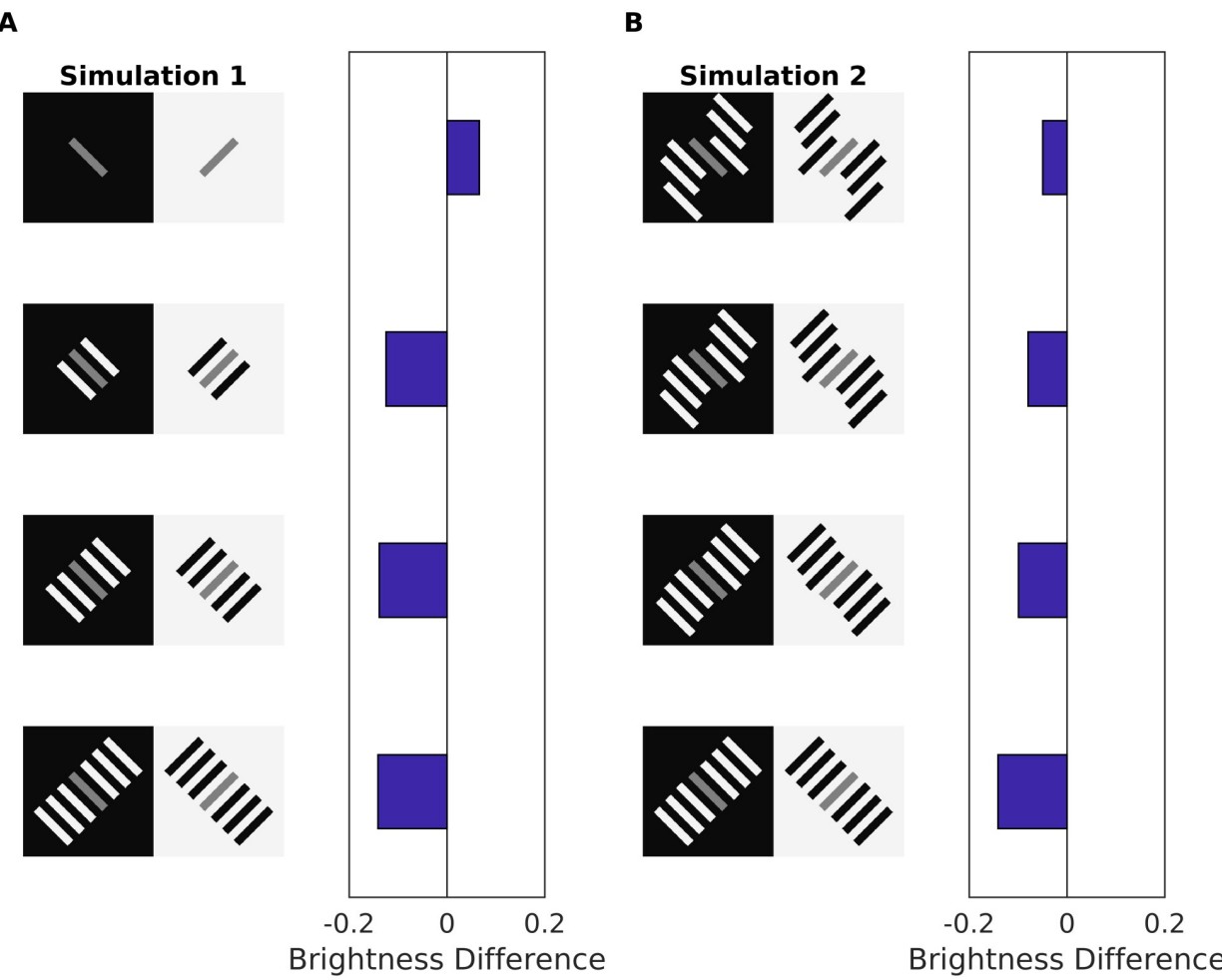

**Fig 12. Model prediction for reverse contrast effect for displays with different configurations.** (A) Reverse contrast with a varying number of adjacent bars to the gray patch. The bar plots show the predicted brightness difference between the gray patches for the corresponding display (a positive value indicates contrast, while negative values "reverse contrast"). (B) Reverse contrast where the good continuation of the end points is varied. This in turn affect the suppression of redundant edges, which increases with the alignment of the flanking bars.

### Brightness assimilation effects

**White's effect.** Fig 13A shows the White's effect, where two gray bars with identical luminance are embedded in alternating black and white stripes. The bar on the black stripe is perceived as brighter as the other one. Lateral inhibition cannot account for this effect, and it has been suggested that the effect is caused by assimilation [66, 93, 94]. Assimilation means that the brightness of the flanking stripes averages with the gray bars, and therefore one expects that reducing the bar height would also reduce the strength of assimilation. However, experimental data indicate that the perceived difference between the bars increases with smaller heights [95], and that bandpass-filtered noise with the same orientation as the stripes enhanced the effect, while with perpendicular orientation the effect was diminished [90]. Therefore, White's effect seems to be principally generated by contrast at the horizontal edges of the bars (Fig 13B and 13C), and to a lesser extend by assimilation from the flanking stripes [96]. In fact, a mainly contrast-based account is supported by the Gain Control Maps of Fig 13A and 13B. Because the vertical edges (assimilation) are highly redundant, their activity is reduced (Scenario 1 & 2). The brightness estimation is dominated by the horizontal edges of the bars

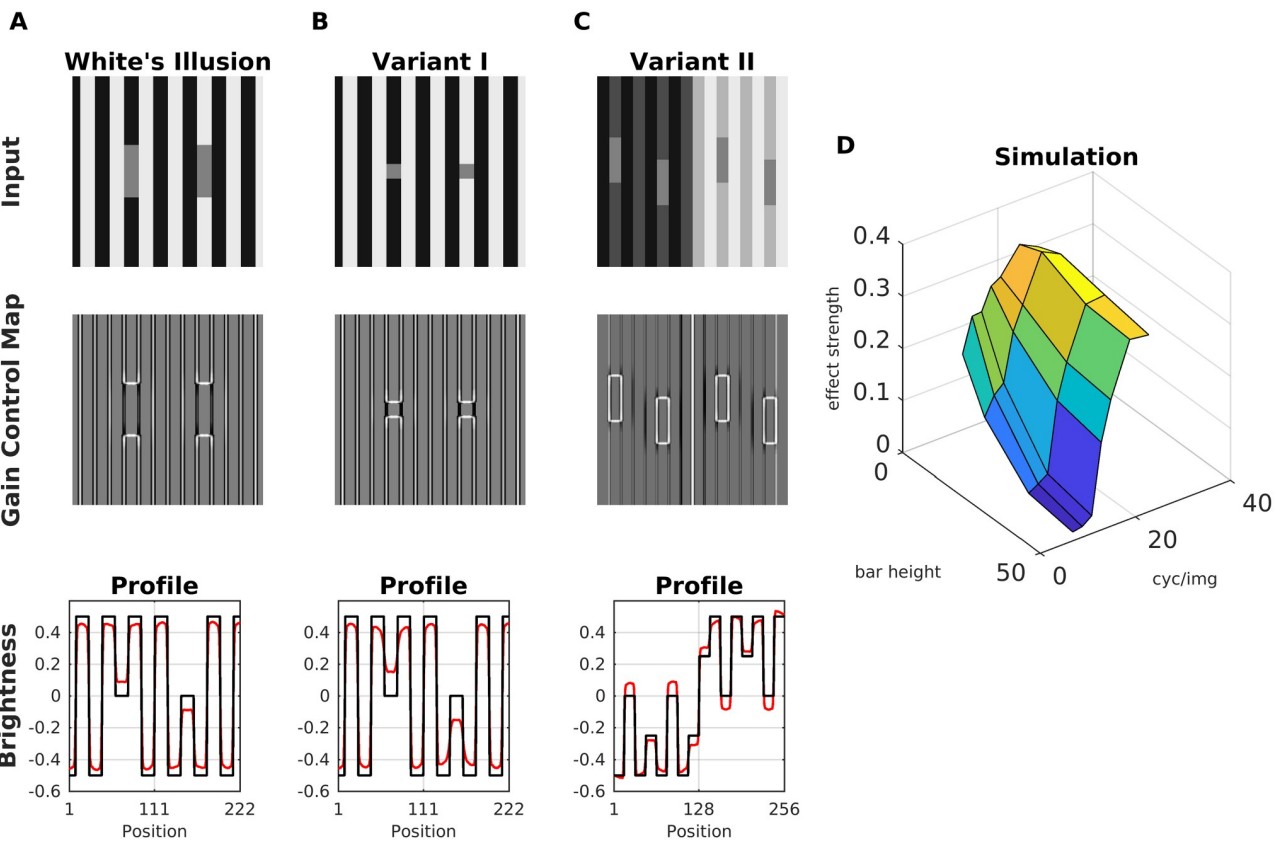

**Fig 13. Model prediction for White's illusion.** (A) Top: White's illusion; middle: the corresponding gain control map; bottom: profile plot of estimated brightness (red line) and luminance (black line). (B) With smaller bar height, the brightness difference between the bars increases. (C) Modification of White's illusion which produces a strong contrast effect. (D) Surface plot of the estimated brightness difference (effect strength) between the bars as a function of bar height (in units of pixels) and spatial frequency of the background stripes (in units of cycles/image). Image size was 256 x 256 pixels.

(contrast), which are enhanced. The vertical edges nevertheless account for a residual assimilation, but the effect on estimated brightness is less for Fig 13B than for Fig 13A. Therefore, the display with the smaller bars (Fig 13B) has a higher predicted brightness difference between the patches, because less activity from the vertical edges "mixes" with that from the horizontal edges during filling-in.

Despite of the presence of stripes (but with low contrast), the display of Fig 13C shows a clear contrast effect of the bars according to Scenario 2. We also studied the relation between the target size and brightness estimation. We observed that the predicted brightness of the bars could be modified as a function of bar height and spatial frequency of the background (Fig 13D). Specifically, the predicted brightness difference between the bars increases both with decreasing bar height and with increasing spatial frequency. These model predictions are in agreement with previous studies [20].

**Todorovic's illusion.** Todorovic's illusion consists of a display with two luminance disks with identical intensity and two sets (black and white) of four squares. The original illusion is designated as Context B in Fig 14, where the disks are occluded by the squares. The test patch occluded by the white squares appears to be brighter than the other. This illusion was originally explained in terms of T-junctions [53]. However, Yazdanbakhsh et al. [50] showed that the effect persisted without T-junctions. Later, [97] studied different variations of Todorovic's

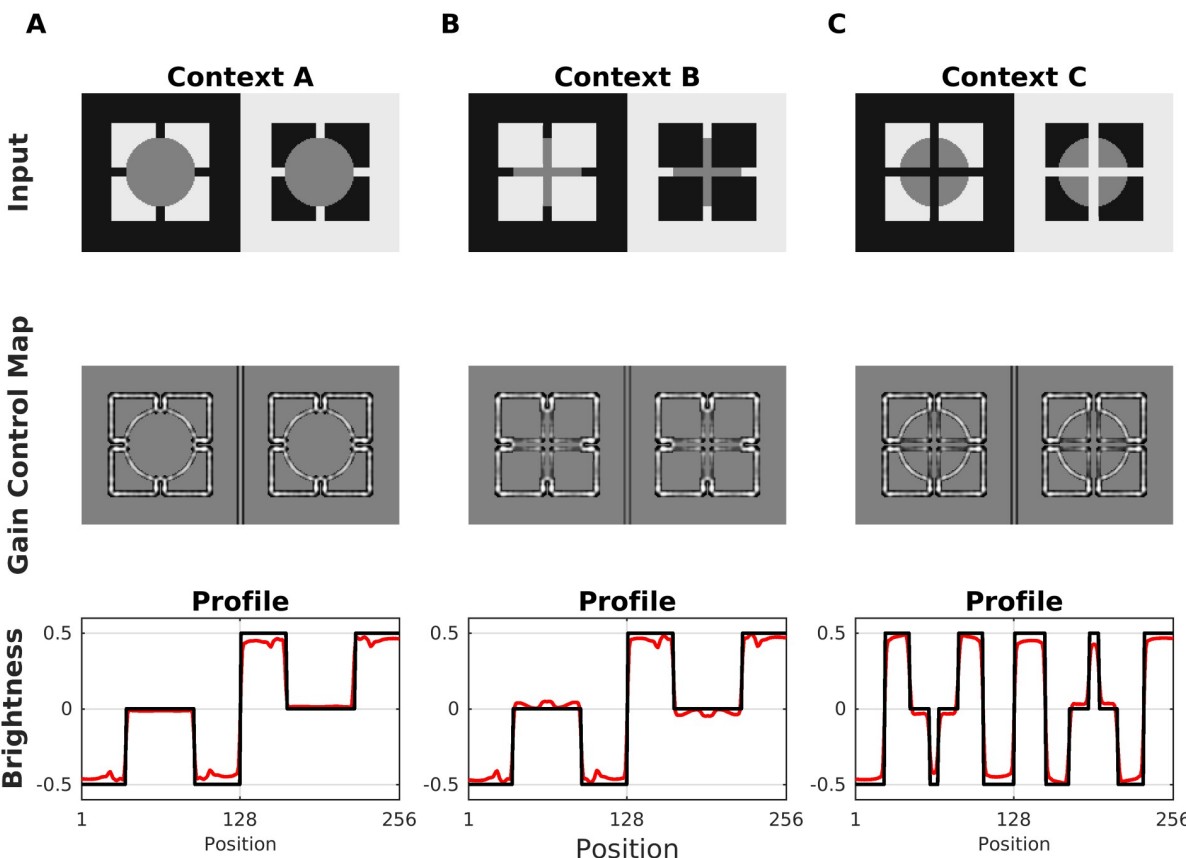

**Fig 14. Model prediction for Todorovic's illusion.** (A) Top: A variation of Todorovic's original illusion with the gray disks in the foreground (Context A). An effect is hardly perceivable. Middle: The corresponding Gain Control Map. Bottom: Profile plots of estimated brightness (red line) and luminance (black line). The model predicted at most a very weak effect. (B) Original Todorovic Illusion (Context B), where the occluded left disk is perceived as being brighter. (C) Reversed Todorovic Illusion (Context C). Now it looks like viewing the disks on a single square background through a window cross, and the left disk is perceived as being darker.

Illusion (labeled Context A and Context C in Fig 14). They found that the target size interacted with the strength of assimilation. The original effect can be reversed according to Context C (Fig 14), which looks like looking through a window cross. It can also be abolished by moving the disk into the foreground (Fig 14, Context A). The bottom row of Fig 14 shows the profile plots of the brightness maps produced by our model. The results could be understood by analyzing the corresponding gain control maps. For Context B, the activity of the occluding edges (disk with squares) is reduced by dynamic filtering. As a consequence, less contrast is produced in the brightness estimation. On the other hand, the edges between the disk and the (black or white) background are enhanced, which produces more contrast in the brightness estimation. This double effect combined to the finally predicted brightness. For Context C, the effect is analogous to Context B, but with opposite disk brightness. With Context A, edge activity along the disks' circumferences is enhanced. Note that the edge covers the squares as well as the background. Although this enhancement of activity produced locally more contrast, it is approximately the same for the two disks, thus producing almost no effect (profile plot of Context B). Fig 15 shows simulation results for the three Contexts as a function of disk size, where we observed qualitatively similar results to previous studies [97].

**More assimilation displays: Dungeon, checkerboard and shevell.** The predictions of our model generalize well to further assimilation displays. Fig 16 shows the Dungeon illusion [52],

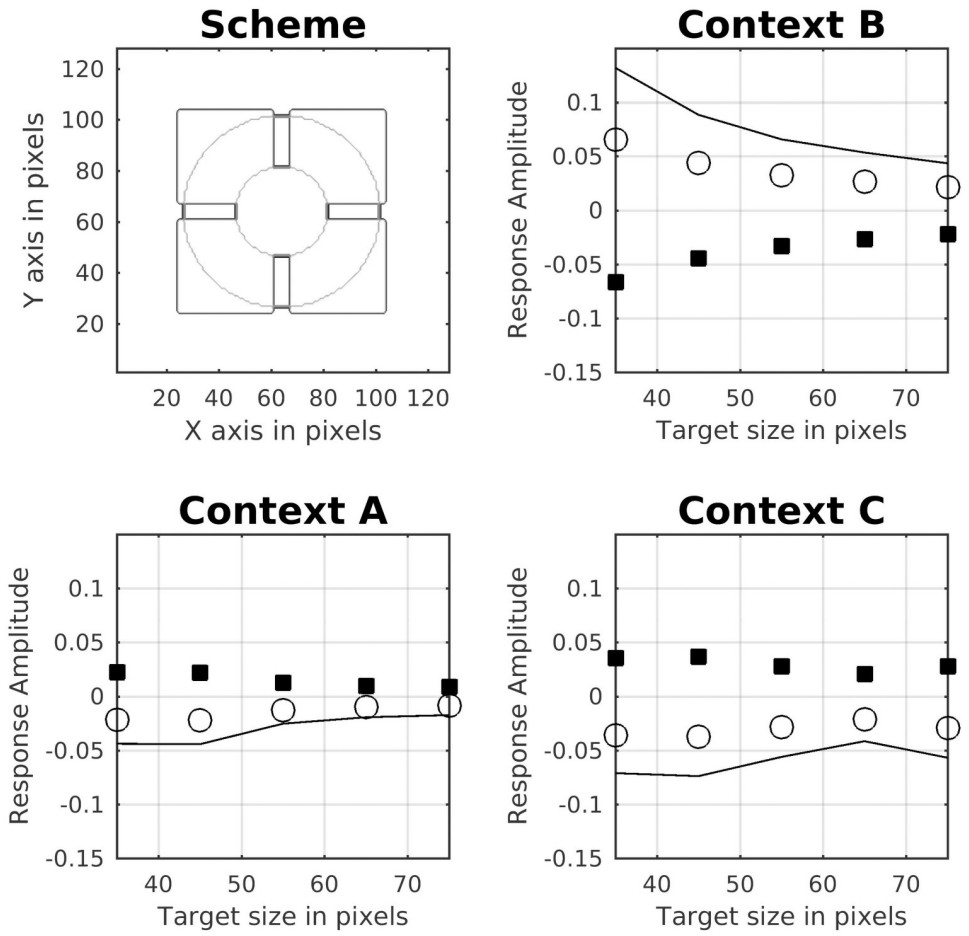

**Fig 15. Brightness dependence on target size for Todorovic's display.** The Scheme shows the smallest and biggest disk size that was used with respect to the squares in order to generate the plots. Each plot indicates the brightness effect for each of the three Contexts shown in Fig 14. The empty circles indicate the predicted brightness of the disk with the white squares. The filled symbols show disk brightness with the black squares. The continuous lines show the estimated brightness difference between both disks and indicate the predicted strength of the illusion.

the Checkerboard illusion [23] and Shevell's Ring [98]. Although all three reveal an assimilation effect on the gray areas, they are different with respect to their spatial configuration. Notice in particular the absence of T-junctions in the Shevell's Ring display. Fig 16 also shows the corresponding simulation results. All three illusions can be explained according to Scenario 2 since the edges corresponding to the gray areas represent redundant patterns with low intensity.

## Further visual illusions

**Craik-O'Brien-Cornsweet effect (COCE).**  Fig 17A shows the COCE, which consists of regions separated by opposing luminance gradients ("cusps") starting at the edges. The cusps drop quickly to a homogeneous gray level and thus the regions between the edges have the same luminance. Nevertheless, especially at low contrast, the gradients seem to fill into the intermediate regions, such that the display is perceived as a low-contrast rectangular wave. The perception of a rectangular wave is less pronounced with high contrast cusps.

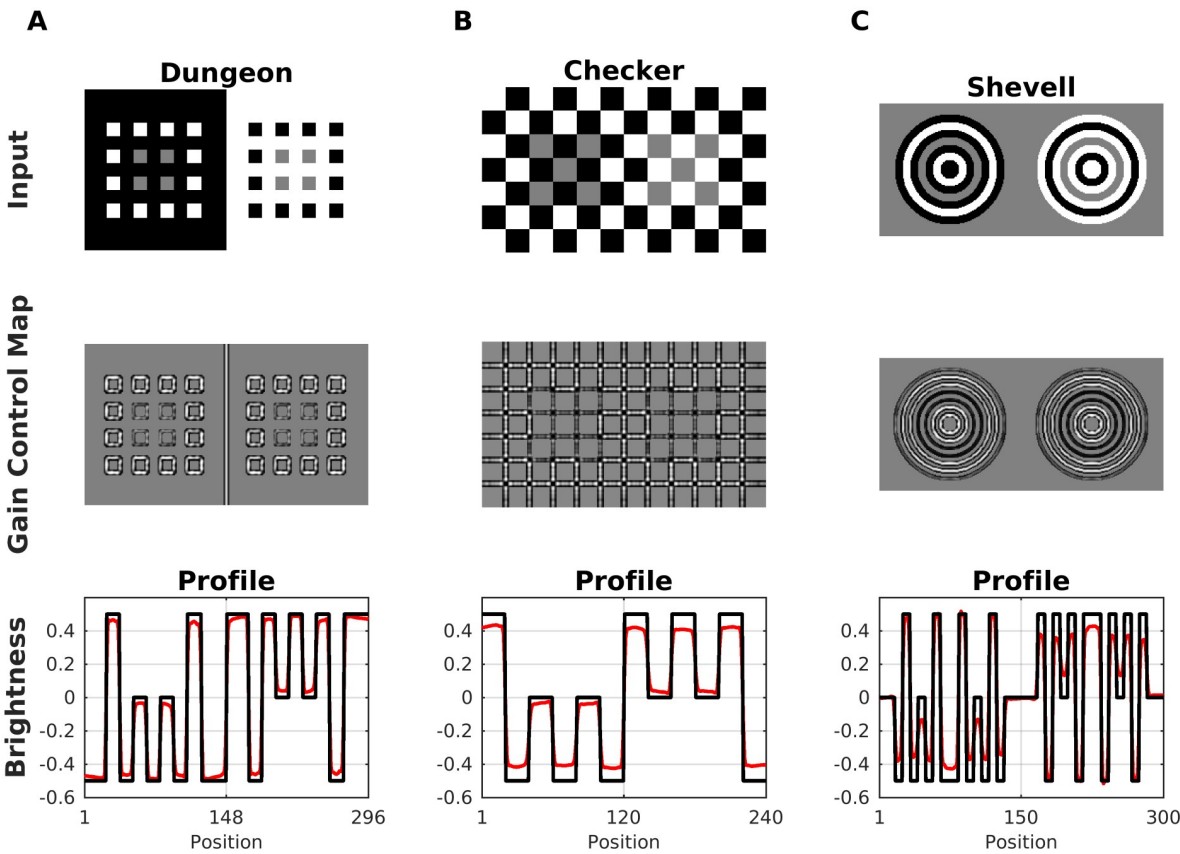

**Fig 16. Model predictions for Dungeon, Checkerboard and Shevell.** In each display, the gray areas have the same luminance, yet they are perceived differently because of assimilation with the adjacent structures. (A) Top: Dungeon illusion. Middle, corresponding gain control map. Bottom, profile plot of the estimated brightness (red line) compared with the input (black line). (B) Checkerboard illusion. (C) Shevell's Rings. Notice that this illusion cannot be explained with T-junctions.

Our model predicted the COCE, but the explanation is more intricate. At first sight, being a filling-in type effect, it should be predicted in a straightforward way by our model. We verified that without gain control (Eq 4), the COCE cannot be predicted. Could it be then that the effect is produced by the low-pass filtering which is applied after the gain control mechanism (see method section). This is not the case, since the removal of low-pass filtering did not affect the prediction of the COCE (see profile plot 2 in Fig 17A). Therefore, the gain control mechanism contributes to producing the effect. Indeed, the luminance gradients cause negative values (indicated by black lines) in the Gain Control Map around the edges (cf. Fig 17A). As a consequence, activity corresponding to the luminance gradients is suppressed by the gain control mechanism, which furthermore reduces the peak activity at the edges. After all, the gradients are "ignored", and our model generates the COCE as a result of assimilation of the edges. This explanation is also consistent with the cow-skin illusion, which is a variant of the COCE without luminance gradients. It is composed exclusively of adjacent black and white lines, and the empty regions are randomly arranged. Fig 17B shows the brightness prediction for the cow-skin illusion.

**Hermann/Hering grid.** Fig 18A (in the top) shows the Hermann/Hering Grid (HG). Although the luminance between the black squares is constant, illusory gray dots appear at the (white) intersections. The textbook explanation deems the center-surround receptive fields of retinal ganglion cells as the principal acting mechanism [99]: Assume a circular receptive field

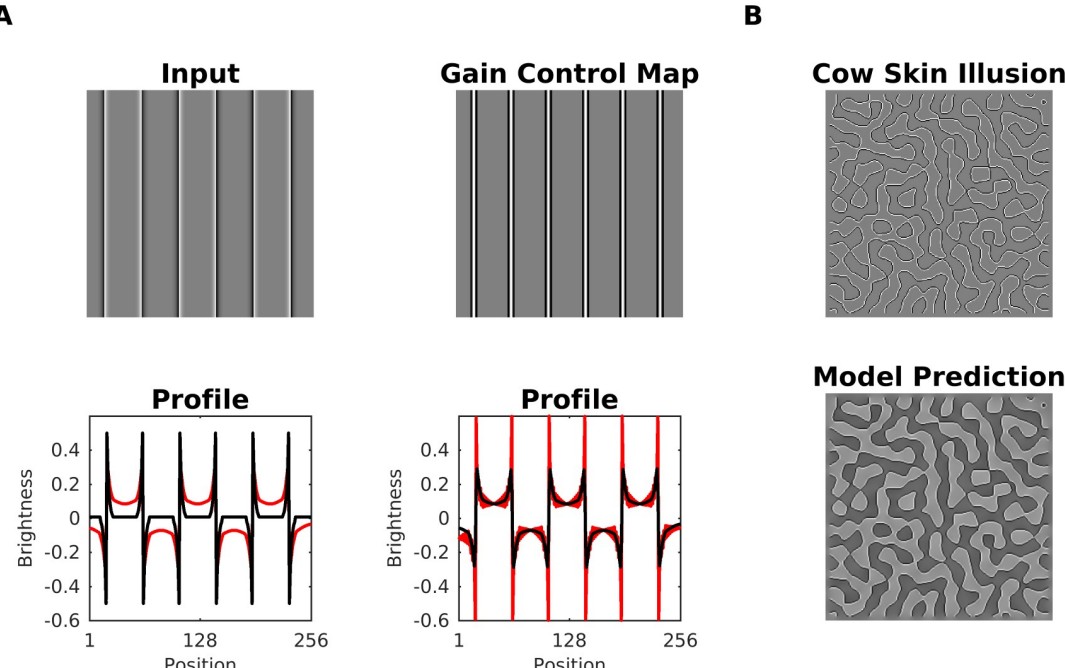

**Fig 17. Prediction for Craik-O'Brien-Cornsweet Effect (COCE).** (A) Top: COCE along with the Gain Control Map. Notice that black lines (adjacent to the edges) in the Gain Control Map, which indicate negative values. Bottom: The first profile plot shows the predicted brightness (red line) along with input luminance (black line). The second profile plot shows the predicted brightness without low-pass filtering after the gain control mechanism (Eq 4). (B) The cow-skin illusion is a variant of the COCE without luminance gradients. It consists only of adjacent black and white lines. Our model consistently predicted this illusion: The brightness map generated by our model is shown at the bottom right.

with an excitatory center which has the same width as the white grid lines. The inhibitory surround covers in addition the black squares. If the center is located right at an intersection, it receives more inhibition from the surround (from two white lines) than when the center is positioned between two intersections (inhibition from one line). This translates to a brightness reduction at the intersections, but not in between. This mechanism, though, is insufficient to explain why the effect is considerably reduced (or even removed) if the bars are slightly corrugated (Fig 18A, bottom). It is also reduced as a function of the ratio between grid line width and block width, where no effect is produced for a ratio of one [100]. Our model predicted the darkening of the intersections for the HG, and the absence of darkening for the corrugated HG. The HG adheres to Scenario 1, where redundant activity is inhibited. Inhibition is especially strong at the intersections of the Gain Control Map. In this way, an assimilation effect is induced. As to the corrugated HG, the corners also represent the redundant patterns, but because the spatial structure are less regular, the inhibition is correspondingly weaker (compare the Gain Control Maps shown in Fig 18B). Consequently, the brightness reduction at the intersections is considerably weaker for the corrugated HG.

Fig 18D shows the dependence of the darkening effect on the ratio between grid line width and block width. In agreement with the results from [100], we find that the darkening effect decreases while the ratio approaches one. We were unable to predict further results with the HG that were presented in [100].

**Luminance staircase and pyramid (Chevreul's illusion).** Chevreul's illusion consists of increasing levels of luminance, arranged as a staircase or as a pyramid. Although luminance is constant at each step, one perceives an illusory brightening on the side of each step where the

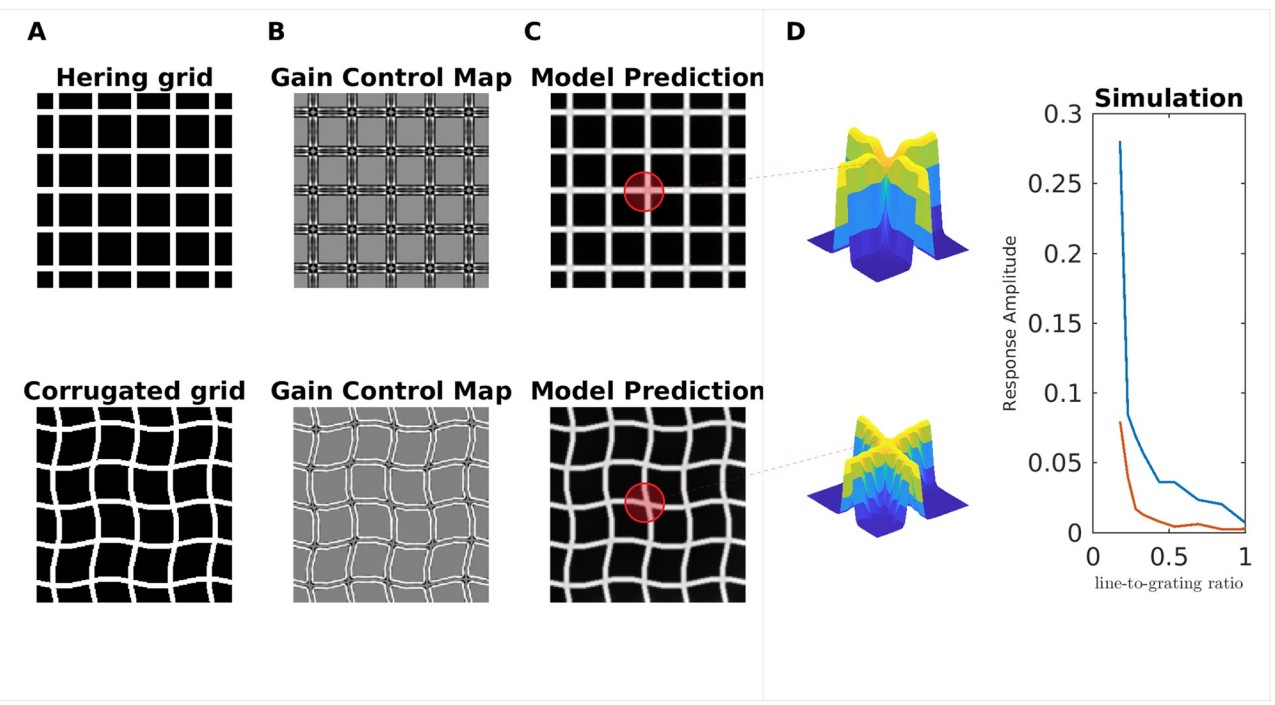

**Fig 18. Model predictions for Hermann/Hering grid and corrugated grid.** (A) Hermann/Hering (HG) illusion and a corrugated version of it. At the intersections of the white grid lines, illusory gray spots are perceived in the HG, but not in the corrugated grid. (B) The corresponding gain control maps of the input images of A. (C) The brightness estimation from our model. The surfaces plots (insets) illustrate the 3D profile of the brightness estimation corresponding to regions highlighted with red. (D) The predicted brightness magnitude at the intersections as a function of the ratio $\frac{\text{line width}}{\text{grating width}}$, where the red curve corresponds to the corrugated grid, and the blue curve to the HG.

adjacent step is darker, and an illusory darkening on the other. In the pyramid version, one perceives in addition (illusory) glowing diagonals (Fig 19). The effect is absent at the lowest (black) and highest (white) luminance level, and is considerably reduced on the middle step for a staircase made up of three steps.

All aspects of Chevreul's illusion are consistently predicted by the gradient system, which is a computational model for representing luminance gradients [10, 101]. The idea behind gradient representations is to capture the smooth variations of luminance (illumination effects) in order to help to disentangle reflectance from the illumination component in luminance (since luminance is the product of reflectance with illumination).

Brightness predictions from our model for the luminance staircase and the pyramid are shown in Fig 19. The illusory whitening and darkening at the stairs can be explained according to Scenario 3: On the one hand, the gain control map increases the activity of the Contrast-Luminance channel. On the other hand, the increase in excitation is offset by the modulation mechanism of Eq 4, thereby producing non-uniform brightness activity at the stairs. The glowing diagonals of the Pyramid Illusion are produced according to Scenario 1, where the activity of non-redundant spatial patterns—especially at the corners—is enhanced. On the other hand, the edges of the staircase represent a redundant pattern, the activity of which is decreased. Consequently, more (less) contrast at the corners (at edges) is generated in the brightness estimation (Fig 19C, bottom). Finally, it is important to emphasize a limitation of our model in this context. We observed that for a big number of steps (i.e., very narrow steps) the dynamic filter "collapses" and the model could not longer predict the illusion nor the glowing diagonals. This is a consequence of the scale-sensitivity of the dynamic filter (i.e., the size of the sampling

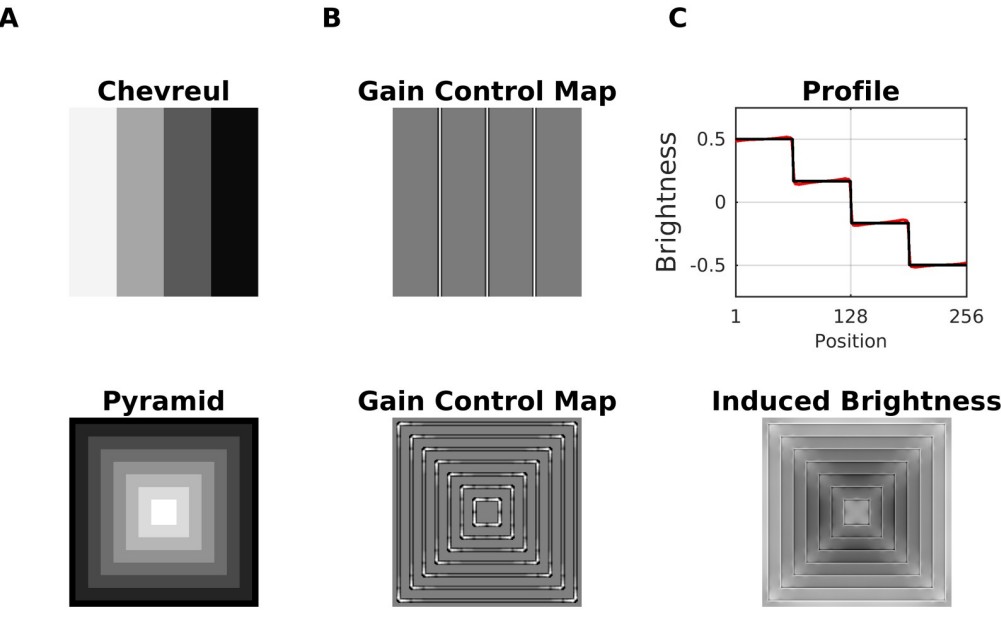

**Fig 19. Model predictions for luminance staircase and pyramid (Chevreul's illusion).** (A) Top: Luminance staircase. Bottom: luminance pyramid. (B) The corresponding gain control maps as a result of dynamic filtering. (C) Top: Profile plot of the estimated brightness (red line) of the luminance staircase (black line). Bottom: The induced brightness, which consists of the difference map between estimated brightness and input luminance (i.e., brighter gray level mean positive values, and darker gray levels mean negative values).

patches), since with decreasing step size, the staircase eventually approaches a linear luminance gradient, and the filter cannot resolve anymore individual steps.

**Mach bands.** Mach Bands [102] are illusory glowing stripes that are perceived adjacent to knee points that are connected with a luminance ramp, where the bright (dark) band is attached to the plateau with high (low) luminance (Fig 20). Notice that Mach bands do not cause Chevreul's illusion. The perceived strength of Mach bands decreases when the ramp gets steeper and eventually approaches a luminance step. Also, for very shallow ramps, the perceived strength decreases. The perceived strength has thus a maximum at intermediate ramp widths [103, 104]. The textbook explanation based on lateral inhibition is insufficient to explain the variation of strength with ramp width—it would wrongly predict maximum perceived strength at a luminance step [101, 105–107]. The perceived strength of Mach bands is also modulated by the proximity, contrast and sharpness of an adjacently placed stimuli [105, 107].

The only computational model published so far that quantitatively predicted all published data about Mach Bands is the gradient system [55, 101]. The gradient system suggests that Mach bands are also perceived at the peaks and troughs of a triangular wave. The gradient system furthermore predicts that bright Mach bands are key for the perception of light-emitting surfaces [10]. Our model predicts the Mach bands, as well as the absence of them at steps (see profile plots in Fig 20). It furthermore succeeds in predicting the inverted-U curve of the perceived strength of Mach Bands as a function of the ramp width (Fig 20C). The inverted-U behavior replicates the trend for measured threshold contrasts for perceiving Mach bands [104]. The threshold is assumed to be minimal where the model predicts the maximum brightness. The measured threshold contrasts for the bright Mach band are also shown in Fig 20C.

The inverted-U curve could be explained by two mechanisms which act in opposite ways. (i) If the ramp width decreases, then the activity at the knee points reaches a maximum that

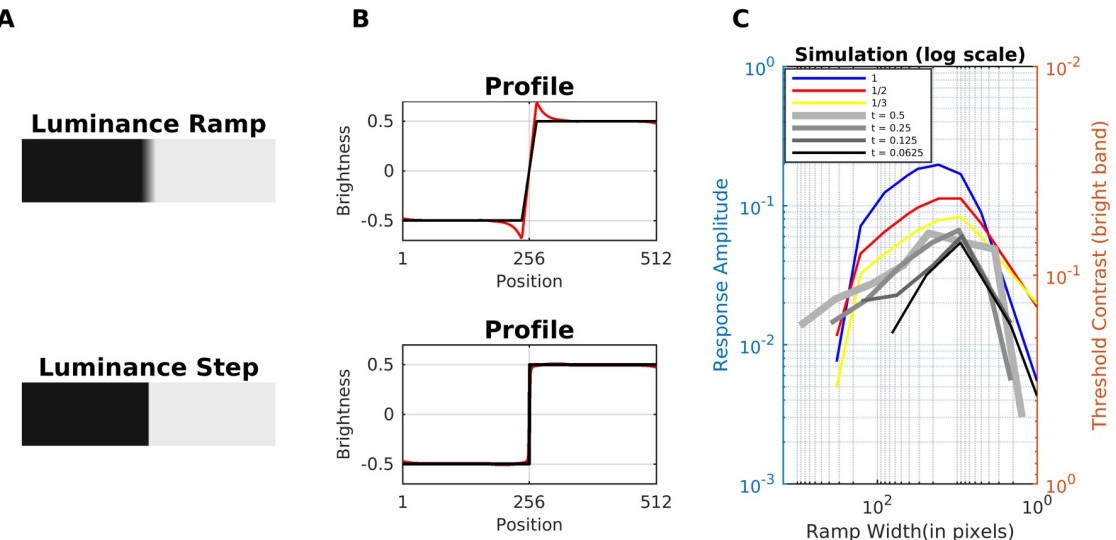

**Fig 20. Model prediction for mach bands.** (A) Top, A luminance ramp that leads to the perception of Mach bands close to the knee points of the ramp. Bottom, a luminance step (no Mach bands are perceived). (B) Profiles plots of estimated brightness (red line) compared with the corresponding input (black line) of A. (C) Brightness magnitude (at the inflection point of the ramp) as a function of ramp width. The plots show the predictions of our model on the perceived strength of the bright Mach band. The colored curves (left axis label: response amplitude) represent model predictions for different dynamic ranges (i.e., differences between luminance values of the upper and the lower plateau, see legend). The gray curves are the threshold contrasts (axis label on the right) for seeing the bright Mach bands at trapezoidal waveforms according to [104]. The trapezoidal waves are characterized by a shape parameter t (see legend; t = 0.5 corresponds to a triangular wave).

renders the modulation mechanism (denominator of Eq 4) of the gain control mechanism ineffective (Scenario 3; ideal step luminance). If the ramp width increases, then the luminance transition between the plateaus is more gradual, which is associated with less activity at the edge locations. In this way, the edge activity becomes more susceptible to the gain control mechanism towards the maximum of the perceived strength. However, this effect does not remain constant. After a certain ramp width, the activity across the ramp gets comparable to the activity at the knee points, which produces less variability in the energy map E. As a result, (ii) the dynamic filter has less effect in Eq 2, reducing gradually the perceived strength induced by the dynamic filtering.

**Grating induction (GI).**    Fig 21 shows the grating induction (GI) display [108], which consists of two sinusoidal gratings (inducers) separated by a gap (test field). Although it has a constant luminance, an illusory brightness modulation is perceived across the test field if the two inducers are in-phase. The phase of the brightness modulation is opposite to the induction wave. The effect decreases when shifting the phase of the inducer gratings relative to each other, being minimum when the gratings are in anti-phase. The illusory modulation is furthermore attenuated with increasing distance between the inducer gratings and with increasing spatial frequency. The GI can be explained in terms of multi-scale filtering [19, 83]), but also by filling-in models [36]. Notice that a common misconception with diffusion-based approaches (=filling-in models) is that the illusory brightness modulation across the test field would average out. This, however, is usually not the case. The exact explanation depends on the model under consideration. For instance, a mechanisms that counteracts "averaging out" are boundary webs from the boundary contour system (BCS) that extend across the test field

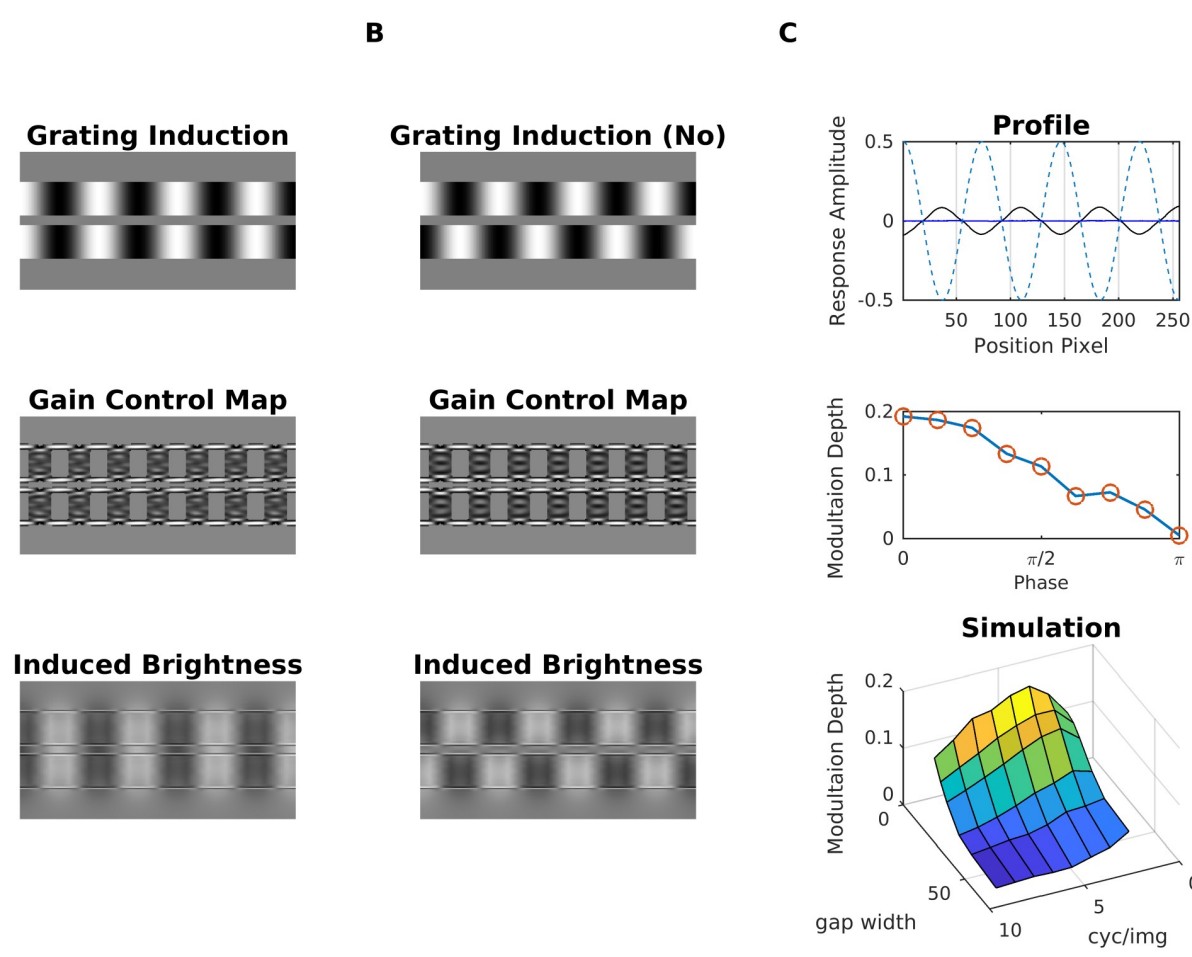

**Fig 21. Model prediction for grating induction.** (A) The Grating Induction refers to the illusory perception of a brightness modulation across the gap (=test field) between the inducer gratings. The brightness modulation is perceived in opposite phase to the inducer gratings. (B) When the inducer gratings stand in opposite phase to each other, then brightness modulation is considerably reduced. The corresponding gain control maps are shown in the middle row, and the last row shows the induced brightness, which consists of the difference map between estimated brightness and input luminance (i.e., brighter gray levels mean positive values, and darker gray levels mean negative values). (C) Top: Profile plot of brightness estimation for display A (black line) and display B (blue line). The dashed blue line shows the luminance profile of the inducer grating A. Middle, modulation depth as a function of the phase difference between the two inducer gratings. Bottom: Surface plot that shows how modulation depth depends on test field width and spatial frequency of the inducer grating.

and trap feature contour activity (FCS) [34]. Other mechanisms include cross-channel inhibition between brightness and darkness activity during filling-in [109].

Fig 21 shows that our model consistently predicted the illusory modulation of brightness across the test field. The strength of the effect decreases in an approximately linear fashion with increasing phase difference between the inducer gratings (Fig 21C). We also observed from a specific spatial frequency on (4 cycles/image) that the brightness modulation decreases with increasing separation and spatial frequency, respectively, of the inducer gratings (surface plot in Fig 21C). Unlike the rest of the illusions the GI effect was produced mainly at the filling-in stage (Eq 5), and to a lesser degree by dynamic filtering. Dynamic filtering increased the activity at the boundaries of the inducer gratings (see gain control maps in Fig 21); this increment produces a significant contrast in estimated brightness between the inducer gratings and the test field, which eventually propagated (by filling-in) across the test field.

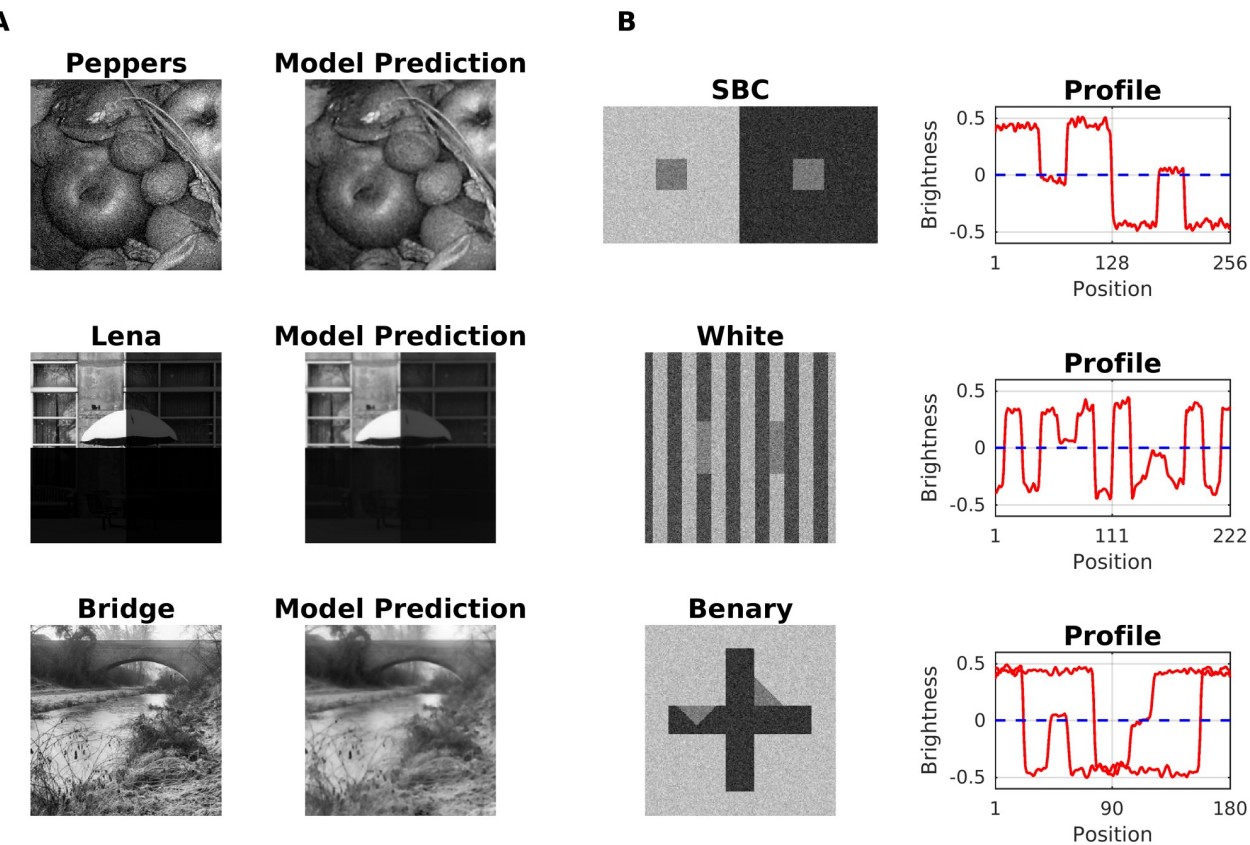

**Fig 22. Real-world image processing.** (A) Top: Fruits image with additive white noise (SNR = 2.6266dB and PSNR = 8.9813dB) along with the corresponding model output (SNR = 5.4682db and PSNR = 11.8228dB); The models capacity for noise removal is worse than an algorithm based on Tikhonov regularization (SNR = 8.30dB and PSNR = 14.65dB; [111]. Middle: A high-dynamic-range version of a real image (where dynamic range of each quadrant decreases clockwise by one order of magnitude) and model output; the dynamic range of the input is 1, and that of the output is 0.9596. Bottom: Bridge image alongside with corresponding model output (SNR = 6.1084dB and PSNR = 13.6201dB). (B) Top: Simultaneous Brightness Contrast display with additive white noise and corresponding brightness profile (red line) as predicted by the model. The dashed line indicates the gray level of the gray squares. Middle: White Effect. Bottom: Benary Cross.

## Real world images and noise

Although synthetic images are a valuable tool for the study of certain aspects of the visual system, it nevertheless evolved to the processing of real-world images. Real-world images provide, therefore, a test of robustness for any model of the visual system. Fig 22 demonstrates that our model is capable of real-world image processing, as well as its ability to handle noisy versions of visual illusion displays (Benary Cross, Simultaneous Brightness Contrast, and White's Illusion). Previously we showed that by using a set derivative filters that cover all orientations (similar to simple cells), Eq 5 globally converges to a stable solution [110]. The convergence is robust against adding noise to the input, or using a high dynamic range of luminance values (Fig 22A, top, middle). Fig 22 suggests that our model's noise reduction performance is worse than a control model based on Tikhonov regularization [111]. The robustness of our model extends to the consistent prediction of visual illusions in the presence of additive noise. In particular, we noted that simultaneous brightness contrast was more sensitive to the presence of uncorrelated noise than assimilation displays. The robustness against noise relates to dynamic filtering, which reduces the correlated (or redundant) spatial information in the edge map. The redundancy of the edges would not be affected by spatially uncorrelated noise. Finally, we

note that dynamic filtering has a couple of limitations with respect to spatially correlated noise, such as band-pass-limited additive noise (see discussion). We did not study this issue in more depth, as it would go beyond the scope of the present paper.

## Discussion

The perceived luminance (brightness) of target structure is highly sensitive to its spatial context. Despite of many modeling attempts for brightness, we still have not arrived at a detailed understanding of the corresponding neuronal information processing principles. With our model, we emphasize the role of decorrelation and response equalization, respectively, in brightness perception. Response equalization is implemented by a dynamic filter that adapts to the spatial structure of luminance patterns in each image in order to reduce the redundancy of boundary maps. Coding strategies that aim at reducing activity and thus energy expenditure in organisms are consistent, for example, with efficient coding [60–62, 64, 112], predictive coding [63], whitening [64] or response equalization [65]. In this sense, we propose that brightness perception is the consequence of suppressing redundant (i.e., predictable) information. Our model is build on the latter idea(s), and apart from being able to process real-world images, it predicts a bigger set of visual illusions than any other previously published model.

Our focus is thereby on low-level vision, as our model simulates the activity of simple and complex cells of the primary visual cortex. For each input, the model learns a filter kernel by identifying redundant patterns in (simulated) complex cell responses (i.e., the edge map), and subsequently uses the filter kernel to suppress redundant information (dynamic filtering). Dynamic filtering amounts to response equalization of simulated complex cell responses, much like the previously proposed "Whitening-by-Diffusion" method which directly acts on the (Fourier) amplitude spectrum [65]. The equalized responses are subsequently used for creating a representation of the sensory input by filling-in (brightness estimation). Nevertheless, dynamic filtering is a global mechanism, which was adopted for the ease of implementation. In the primary visual cortex, we expect that dynamic filtering acts in a more local fashion, but still on a spatial scale that exceeds the typical receptive field sizes of V1 neurons. Such non-local mechanisms could be biologically implemented by feedback from mid-level visual neurons with sufficiently big receptive fields for detecting non-local correlations in activity.

We believe that our success in predicting a relatively large number of visual illusions lends some support to our proposed computational principle. Without changing any of our model's parameter values, we are able to predict Simultaneous Brightness Contrast (SBC), White's Effect, Reverse Contrast, Benary's Cross, Todorovic's illusion (with variations), the Dungeon Illusion, the Checkerboard Illusion, Shevell's Ring, the Craik-O'Brien-Cornsweet effect (COCE), the Hermann/Hering grid, the corrugated grid, Chevreul's illusion (including the luminance pyramid), Grating Induction (GI), and Mach Bands. Additionally, for some of the illusions, we were able to reproduce the trend for corresponding psychophysical data (SBC, White, Reverse Contrast, Hermann/Hering grid, Todorovic, GI, and Mach Bands). Despite all of these successes, we must not forget to mention some of the limitations of our model. We cannot predict illusions—without modifying the current parameters—such as achromatic neon "color" spreading, the Ehrenstein illusion, Chubb's Illusion and some variations of the Hermann/Hering grid; Reverse Contrast with different grouping factors, SBC with articulated noise, and Mach Bands with an adjacently placed stimuli.

A further limitation is handling visual illusions where a target patch is surrounded by an articulation pattern. An articulation pattern can be created from a region with uniform luminance. The region is subdivided into small square patches. The luminance of each patch is randomly modulated according to a Gaussian random variable, with the mean value being the

original luminance value, and the standard deviation being the modulation depth. The average luminance across the articulated pattern has to be identical to the luminance of the original uniform region. The articulation patterns would introduce additional spatial redundancy into a luminance display, and the kernel would eventually learn this excess redundancy for dynamic filtering. As a consequence, dynamic filtering may modify the target's edge representation in an unpredictable way. For brightness estimation, this could mean, for instance, that assimilation effects turn into contrast effects or vice versa. This behavior appears to be inconsistent with current psychophysical observations [113], and may hint at additional mechanisms that need to be considered. It cannot be ruled out that additional mechanisms reduce the redundancy along other stimulus dimensions as well, for example luminance, relative contrast, or auto-correlation. The global nature of the dynamic filter represents another trade-off. In order to learn the kernel for dynamic filtering, we sample patches randomly across the input. As a consequence, local statistical information between (unrelated) patches that are far away from each other could be intertwined, affecting their brightness predictions. A possible solution would be to introduce local constrains upon sampling, or as well to introduce a local normalization function, which takes into account local spatial auto-correlations.

### Comparison with other models that predict contrast and assimilation

Our approach produces contrast effects by enhancing non-redundant edges through dynamic filtering according to Scenario 1. Assimilation effects are generated by suppressing redundant edges as a function of their relative intensity with respect to the other edges (Scenario 2). We thus do not make any explicit assumptions about how a visual target is related to its context in terms of segmentation, or belongingness and perceptual frameworks, respectively. We do not require the categorization of image features either. In this sense, our approach is more general than previous computational proposals and theories [47, 49, 114], which purport that a stimulus is divided into perceptual frameworks based on anchors [114] or T-junctions [49]. However, it is not clear whether anchors or T-junctions are sufficiently robust cues in real-world images, and actually few previously published models demonstrated the processing of real-world images.

Dynamic filtering is sensitive to the correlation structure of spatial patterns in order to generate contrast and assimilation effects. In this way, the output of our model would not be significantly affected if uncorrelated noise was added to the input. Yet multi-scale models are highly sensitive to additive noise [19, 20, 22], because their predictions depend on a careful readjustment of filter responses across spatial frequencies. Thus, if noise was added to contrast and assimilation displays, then corresponding predictions would be altered, because of corresponding changes in the spatial frequency spectrum [24].

Our model adapts to the statistical structure of each input image. This is to say that we do not evaluate each input image in a previously learned long-term statistical context. A long-term statistical context usually is learned from a big number of input samples in order to derive feature-specific probability distributions. In connection with brightness, a relationship between occurrence frequency of certain types of natural images and brightness perception has been proposed [56–58]. The main limitation of such models is that they require an enormous amount of data, and that visual illusions act much like an associative trigger or they are perceived according to humdrum Bayesian inference.

### Conclusion

One might ask whether the range of illusions that we successfully predict with our model can be attributed to a common mechanism. The answer is yes, and the underlying mechanism is

dynamic filtering. Dynamic filtering acts to equalize the amplitude spectrum of a boundary map. In the spatial domain, thus, dynamic filtering depends on pattern redundancy (but not on activity). In this way, non-redundant patterns are enhanced (i.e., contrasted) compared to to redundant patterns (which are assimilated). In conclusion, this study provides a proof of concept of a hypothetical information processing strategy for visual system, based on economizing edge representations. Our predictions are reliant on the self-structure of the visual input, but not on accumulated visual experience, spatial frequency representations, or predefined detectors. Our proposed mechanism does not exclude information processing principles like accumulating visual experience or spatial frequency representations, and should be considered as being complementary to these. Finally, future work should address the understanding of how the statistical structure of the context surrounding a target patch influences its appearance. We also plan to study how different noise structures (as narrow-band, oriented, or correlated) influences the predictions of our model. Our redundancy-reduction hypothesis should be compatible with all levels of information processing. This means that redundancy reduction likely might apply to higher-order patterns and shapes that form the primitives for object recognition.

## Supporting information

**S1 Text. A. Gabor filters**. In this section are described the parameter values and a mathematical description of unbalancing the ON/OFF subregions for the filters used in the Contrast-Luminance channel and the Contrast-only channel. **B. Energy Map**. In this section are included the mathematical details corresponding to the local energy map. **C. Dynamic filtering with zero-phase whitening (ZCA)**. In this section are described the mathematichal details to perform dynamic filtering of our model. **D. Solving** Eq 5. In this section, a solution for Eq 5 is derived, which is used to estimate the output (brightness map) of our model.
(PDF)

## Acknowledgments

We thank the Visca Group for laboratory space and Gonçalo Correia for useful discussions.

## Author Contributions

**Conceptualization:** Alejandro Lerer, Matthias S. Keil.

**Formal analysis:** Alejandro Lerer, Matthias S. Keil.

**Investigation:** Alejandro Lerer, Matthias S. Keil.

**Methodology:** Alejandro Lerer.

**Project administration:** Hans Supèr.

**Resources:** Hans Supèr.

**Software:** Alejandro Lerer.

**Supervision:** Hans Supèr, Matthias S. Keil.

**Validation:** Hans Supèr, Matthias S. Keil.

**Writing – original draft:** Alejandro Lerer, Matthias S. Keil.

**Writing – review & editing:** Alejandro Lerer, Matthias S. Keil.

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
