## [Decision Letter · Decision Letter 0]

18 Aug 2020

Dear Alejandro et al. -

Thank you very much for submitting your manuscript "Predictive coding as a unifying principle for explaining a broad range of brightness phenomena" for consideration at PLOS Computational Biology.

Your manuscript has been carefully reviewed by members of the Editorial Board and by three experts in the fields of brightness/lightness perception and computational neuroscience. The reviewers all agree that this is an impressive piece of work that addresses an exceptional range of perceptual phenomena. In light of the reviews (below this email), we would like to invite the resubmission of a significantly-revised version that takes into account the reviewers' comments. Please provide a point by point response to their criticisms with your revised manuscript.

In addition to the points raised be the reviewers, I have a few points of my own that I would like to see addressed in your revision. First, the proposed connection between response normalization and predictive coding is not spelled out in your paper as clearly as I think it should be, especially given that the phrase "predictive coding" appears in your manuscript title. As Reviewer 1 pointed out in his/her review, many current brightness models already invoke response normalization. But, to my knowledge, a specific connection between response normalization and predictive coding has not previously been proposed in the context of brightness models. This is a nice innovation, but your manuscript does not really explain what you mean by it. Instead, you let the reader draw their own conclusions about the implications of a predictive coding mechanism for brightness.

I usually think of predictive coding as a neural feedback mechanism that results in the encoding of an error signal. It is not clear to me how such a mechanism would perform an ecological function in the context of brightness perception. Are you suggesting that what we consciously perceive when we look at the world is an error signal—or do you mean something else? Does response normalization in brightness instead perform a function that is less "severe" than that? If so, why does this mechanism deserve to be called "predictive coding?"

Second, as noted by Reviewer 1, my own work on lightness/brightness modeling also combines edge integration with a contrast gain control mechanism (see for example, my Journal of Electronic Imaging article cited by Reviewer 1). I have used my model to fit only a much smaller range of brightness phenomena than that which you address here. Nevertheless, the data that I have addressed seems to place very strong mathematical constraints on any model that might be proposed to account for them. Most importantly in the current context, I don’t think my data can be easily understood in terms of response normalization. I’d like you to address this in your paper. Can your model account for my results? If so, please explain how. If not, then you should acknowledge the existence of an alternative model that also combines edge integration with contrast gain control and just say that it is not clear if the two models are compatible.

We cannot make any decision about publication until we have seen the revised manuscript and your response to the reviewers' comments. Your revised manuscript is also likely to be sent to reviewers for further evaluation.

Sincerely,

Michael Rudd

Guest Editor

PLOS Computational Biology

Lyle Graham

Deputy Editor

PLOS Computational Biology

Dear Alejandro et al. -

Your manuscript has been carefully reviewed by three experts in the fields of brightness/lightness perception and computational neuroscience. The reviewers all agree that this is an impressive piece of work that addresses an exceptional range of perceptual phenomena. However, the reviewers have also raised a large number of specific points that will need to be addressed before a final decision can be made regarding publication. Please provide a point by point response to their criticisms with your revised manuscript. In addition to the points raised be the reviewers, I have a few points of my own that I would like to see addressed in your revision. First, the proposed connection between response normalization and predictive coding is not spelled out in your paper as clearly as I think it should be, especially given that predictive coding appears in your manuscript title. As Reviewer 1 pointed out in his/her review, many current brightness models already invoke response normalization. But to my knowledge, a specific connection between response normalization and predictive coding has not previously been proposed in the context of brightness models. This is a nice innovation, but your manuscript does not really explain what you mean by it. Instead, you let the reader draw their own conclusions about the implications of a predictive coding mechanism for brightness. I usually think of predictive coding as a neural feedback mechanism that results in the encoding of an error signal. It is not clear to me how such a mechanism would perform an ecological function in the context of brightness perception. Are you suggesting that what we consciously perceive when we look at the world is an error signal—or do you mean something else? Does response normalization in brightness instead perform a function that is less "radical" than that? If so, why does this mechanism deserve to be called "predictive coding?" Second, as noted by Reviewer 1, my own work on lightness/brightness modeling also combines edge integration with a contrast gain control mechanism (see for example, my Journal of Electronic Imaging article cited by Reviewer 1). I have used my model to fit only a much smaller range of brightness phenomena than that which you address here. Nevertheless, the data that I have addressed seems to place very strong mathematical constraints on any model that might be proposed to account for them. Most importantly in the current context, I don’t think my data can be easily understood in terms of response normalization. I’d like you to address this in your paper. Can your model account for my results? If so, please explain how. If not, then you should acknowledge the existence of an alternative model that also combines edge integration with contrast gain control and just say that it is not clear if the two models are compatible.

Reviewer's Responses to Questions

**Comments to the Authors:**

Reviewer #1: The manuscript describes a new version of an edge-integration model of brightness perception and shows that it gives a good account of a wide range of brightness/phenomena/illusions. Edge-integration, or “filling-in” models of brightness/lightness perception have a long history in visual science, beginning with the Retinex model of Land & McCann (cited) and more recently in a series of studies by Michael Rudd, none of which curiously are cited or discussed in the present article. The proposed model variant emphasises the incorporation of predictive (or probably better termed efficient) coding, specifically filter response equalization, and the title of the manuscript suggests that this is the new unique feature of the model. However it is not clear to this reviewer in what way it is new given that many existing models of brightness perception incorporate contrast/response normalization in their model architectures. Having said that the model’s predictions of a range of brightness illusions are well-described and convincing. I was particularly impressed by the ability of the model to predict the disappearance of Hermann grid spots in a corrugated grid. Whether the proposed model variant of edge-integration/filling-in will have any impact on a field that is already replete with models of brightness phenomena is hard to say, but I see no reason why the model should not be published. All I can do here is to suggest some possible ways to make the manuscript more focused and digestible.

1. Although it is commendable that in their Introduction the authors attempt to summarise most of the existing approaches to studying brightness phenomena it is perhaps too tall an order to attempt to do this in a non-review article. The authors in any case omit to mention some important approaches, for example the “Intrinsic-image” approach best associated with the work of Adelson/Anderson/Ekroll/Logvinenko and others, and the “Feature model” approach best associated with Watt/Morgan/Morrone/Burr/Georgeson and others. As a suggestion it might be better to alert the reader in the Introduction to sources that provide details of existing approaches to brightness/lightness perception (e.g. Gilchrist’s book “Seeing Black and White”; Kingdom’s 2011 review article in Vision Research), and concentrate instead on providing the rationale for the edge-integration approach in the context of its many existing model variants and the critical empirical evidence that speaks to its validity.

2. The main pitch is “predictive coding”. This appears to be implemented in the proposed model by selective enhancement of edges via response equalization. Yet response equalization is a feature of many existing models of brightness perception. In the case of multiscale filtering models it is a critical feature of, for example, Blakeslee & McCourt’s ODOG model (cited), Dakin & Bex’s model of the Cornsweet Illusion (not cited: Proc. Roy. Soc. B., 2003, 270) and Kingdom’s model of Mach Bands (cited). In the case of edge-integration models contrast gain control is a feature that runs throughout the models of Rudd & colleagues (e.g. Rudd, JOV, 2010, 10(14):40; Rudd, JOV, 2013, 13(14):18; Rudd, J. Electronic Imaging, 2017, 26(3)), though whether the contrast gain adjustments in Rudd’s models constitute examples of predictive coding is debatable. If the authors want to make the case that predictive coding is a novel feature of their model they need to explain in what way it is novel given existing models of brightness perception that also incorporate response/contrast equalization.

3. A critical property of the proposed model is “filling-in”. There has certainly been some recent evidence that favors the filling-in approach over multi-scale filtering models, for example the study by Betz et al. (cited). It would help the reader if this evidence was flushed out with a bit more detail in the Introduction as part of the rationale for adopting an approach that is still highly controversial.

4. The authors might want to speculate as to the function of the low-spatial-frequency filters in the cortex that play no part in their model of brightness perception, i.e. those that are lower in spatial frequency than the “contrast-luminance” channel in their model. A related issue worth commenting upon is this: if brightness perception is so dependent on the presence of edges detected by high-resolution filters, why are brightness illusions often enhanced by lowpass filtering (e.g. grating induction – see McCourt & Blakeslee, 1993, Vis. Res., 33, No. 17; White’s Effect – try it yourselves!)?

5. Although the “boundary contour system (BCS)” and “feature contour system (FCS)” referred to by the authors are initially presented as hypothesised model features, they rapidly morph in the text to the status of established facts. They are not facts and should only be referred to as possible mechanisms.

6. Page 2 “Typically, multi-scale models adjust the response amplitudes….in a way that follows the shape of the contrast sensitivity function”. If by contrast sensitivity function you are referring to the threshold version this statement is both incorrect and a trivialisation of multi-scale models, at least for the versions mentioned in point 2 above. Best to check the details of these models before making this claim.

7. A key feature in the proposed model is the use of just two scales of filtering: high resolution “contrast-only” and low resolution “contrast-luminance” filters, the latter instantiated by filters that are not dc-balanced. First, what are the peak spatial-frequencies of the two types of channel expressed in cycles-per-image, and hence what is their spatial-frequency ratio? The answer to this question will help the reader understand the range of filter scales that have been omitted from the model computations, and thus be in a position to make up their own mind as to whether they consider this to be realistic. Second, in the proposed model, the spreading of neural activity in the filling-in process appears to be instantiated by the low-resolution contrast-luminance channel, yet even after multiple readings I cannot fathom what exactly is the role of the contrast-only filters in the filling-in process. Do they act as borders to gate the extent of filling-in, or what?

8. Stylistic. I have not come across the approach of using a reference number as the subject of a sentence or clause, e.g. “More recently, [28] extended the mechanism...” Is this a style recommended by PLoS? If not this needs fixing throughout. Also, the introduction of author names without prior referencing is very confusing, e.g. “Thus, Domijan modified etc.”

Reviewer #2: Authors present a " a hypothetical information processing strategy for visual system" (L726) which, from a raw visual input, outputs an estimate of a perceived brightness value. This model is then applied to different brightness illusions and fitted to their qualitative effects. While the paper and results are interesting, I have some points of concerns which necessitate a major revision. I am first concerned by the clarity of the presentation. For instance the introduction uses terms which may not be customary to the readership of PLoS CB but more to vision neuroscientists (eg L53 Eigengrau, L78 Benary cross). A synthetic figure which would present the different brightness illusion displays that are used in the text would be better justifying the point of the paper and simplifying the presentation of the introduction. A similar strategy applies to the rest of the manuscript (+ check syntax, eg beware of using mixed verb tenses as is the case in the author's summary).

My biggest concern though is scientific. How do you justify the derivation of your model? There are lots of phenomenological models like yours, and some are dealing with the same problem that you try to solve. For instance, you may check recent papers by Dario Prandi (eg https://journals.physiology.org/doi/full/10.1152/jn.00488.2019 ). First, there should be a justification for the choice of some parameters (eg threshold (L268) and sigmoid (L270) ...) or at least describe what would happen if you change these. More than that, you claim in the conclusion that you have principled derivation of your model based on predictive coding, but I do not see that clearly enough in the Methods section. I see two parts: one encoding the image into a compressed edge-like representation and regularized using some prior knowledge, then an iterative reconstruction (decoding?) inducing the brightness illusion. See for instance work from Spratling for similar architectures. This would be a great contribution of your paper.

I have some other points:

- embed your figures within the manuscript. It is tedious to go back and forth.

- Figures appear at a low quality, especially figure 1. Use a vectorial format like PDF or EPS.

- how do you justify physiologically and mathematically the existence of the two pathways (contrast + luminance) ?

- you use flat, rectangular coordinates, not a retinotopic log-polar mapping. Specifically, you would not have an effect of scale as your Gabor filters have only one scale. For instance, the brightness illusion in the Hering grid (fig 16) is greater in the periphery - could you justify that?

- Energy map, S2 : from Parseval theorem, the local energy in Eq 10 is the sum of the spectral energy of each Gabor, and thus that from a DoG ?

-figure 12 context B: I would tend to see a convex object occluded by the square - could this explain the illusion?

-figure 20 : if you do a quantitative measurement, you should compare it with some other denoising model as a control. you do not need a 4 digit precision.

minor:

-------

L176: missing closing parenthesis

L209 : regarding the choice of Gabor filters, these contain a DC component - could you think of alternate filters that would better match to V1 cells?

L224, l248, ... : LaTeX parenthesis `` or '' are mixed up. repeats throughout the text

L269 "sigmoidal function" > "sigmoid function"

L277 missing variable name in "The value of XXX acts "

caption Fig4 "CL" > "LC"

caption Fig9 "Gain Control Maps" < "Gating Maps"

caption Fig19 "Garing" > "Grating"

Reviewer #3: Overview

This is a bold and interesting paper which presents a new model for the encoding of image brightness in perception, using a few broad, general computational principles aimed at reducing redundancy in boundary maps (p.20). The paper is bold because it attempts to account for a very wide range of brightness effects ('illusions') that have been described over a century or more of research. A long-standing challenge has been to account for 3 kinds of effects (contrast, assimilation, and filling-in) within a coherent framework. The model does a surprisingly good job of this, but the authors are also clear in describing where the model does not work, with some useful speculations in the Discussion about what else might be needed.

The paper covers a lot of ground, but is pretty compact in length. The Introduction is particularly useful in summarising both the (large) range of brightness effects studied in the literature, and the great variety of ideas that have been proposed to account for these effects.

The paper does well in presenting a verbal description of the model and its several stages, while reserving the mathematical details for the Appendix. Even so, I (not a computer scientist by trade) struggled to grasp the fillling-in process (Stage 3, 'brightness estimation'). In particular I could not see how, in the iterative process, the brightness image (z) got started, or what was being adjusted to reach a final result. Anything that would make this clearer to the reader would be valuable.

The paper is well-organized and generally clear, but English expression in the paper does not always flow as well as it might, so the paper would benefit from a 'light-touch' editing job by a native English speaker. I have made (below) some suggestions for syntax improvement, but more is needed.

Detailed commentary

References: Many references lack full information about volume & page numbers. Need checking.

L. 133, L. 371, L. 388, L. 768, L. 797: 'what' >>> 'that'. It would be best to check all instances of 'what'

L 164: 'that' >>> 'which'

L 224: as well as defining what 'g' is, state that Im(x,y) is the input luminance image (if this is so).

L. 264-271. Some reason and motivation for the various steps in this 'Gain Control Map' section would be helpful. And similarly for the subsequent section on "Gain control and normlization...".

It would also useful to know where the chosen parameters (eg a, b, tau, filter parameters, alpha, threshold 'theta', etc) came from, and whether their values were critical.

L. 277: Something missing here: "The value of acts as an upper bound to the maximum activity"

L. 278-9: 'bigger (or smaller) than zero' >>> 'greater (or less) than zero'

Fig. 5 and later: black & blue are a poor choice of colours for the 'profile' curves in these figures; not very distinctive at all. My suggestion would be: black for input luminance, red for model brightness response. Plot the red first, then the black; that way the model's (red) deviations from luminance will appear very salient, as red segments.

L. 326-7: you say "In this case, the major contribution to predicted brightness comes from the Gain Control Map (numerator of Eq 4)." But this seems very unclear to me, because the G(x,y) term appears in both the numerator & denominator; so some further explanation is required, as to why Eq 4 has the form it does, and what it achieves.

Fig. 6: The input images (left column) clearly need a thin black outline to mark their full extent. At present, the upper input image apears to be a black square, rather than a white-black edge; and the lower image appears to have 3 regions, but actually has four.

L. 366-368. On SBC, there appears to be a direct contradiction here: You say that "psychophysical studies report that the contrast effect is perceived less intense for smaller patches [60-64]", but Fig. 7D appears to show the weaker contrast effects at larger target sizes (not smaller), for both data & model. Something wrong here ?

[You may perhaps be creating some confusion by switching between the use of "brightness" and "contrast" terms too often ? And since the brightness values in Fig 7B can be both +ve & -ve, but those in panel D are purely positive, it's unclear what is being plotted in panel D: the 'brightness' of the right-hand patch, or the 'contrast' (ie difference ?) between left & right patches ? Or what ?]

Also, what is the published source for data in Fig. 7D ? Or are the 'data' actually model responses, not experimental data ?

Fig 7, headings: Gating Map >>> Gain Map ?

Fig 10, axis labels: "Brigthness" >>> "Brightness"

L. 426: 'to a less detect' >>> 'to a lesser extent' ?

Fig 12, caption, last line: 'left square' >>> 'left disc' ?

L.657: "is dynamic filtering a global mechanism" >>> "dynamic filtering is a global mechanism"

L. 663-671: Yes, a major strength is the range and variety of effects that can be reproduced by this model. And you also say that "for some of the illusions, we were able to replicate data from psychophysical experiments". To support this further, it would be good to see, for a few cases, actual comparisons between published experimental data and the model's predictions. Is that possible ? I realize that there might be a problem in expressing the model and data in the same metric, but perhaps there are ways around that ?

Also, you mention that for all the cases examined here the same model parameters were used throughout. That is important, and could be mentioned much earlier in the paper.

L. 679: define 'articulation pattern'

----------------

Appendix, L. 740-744. Eqn 7 contains undefined terms x_hat, y_hat. And orientation 'theta' does not actually appear in the equation. Earlier (Eqn 2 & L.268) 'theta' appears as a threshold; this duplication would be best avoided by finding another symbol for the threshold.

Appendix, L. 748-752 are puzzling. First, gON and gOFF are not actually defined, though presumably they are the +ve and -ve regions of the Gabor RF, g, in Eqn 7. Second, and more importantly, the motivation for normalizing these 2 sub-fields separately is not described, and my guess is that it will have a different impact on the odd and even members of the Gabor pair. Does it ? And because of this nonlinearity I think the pair of RFs are not actually a Gabor pair any more, and neither (strictly) is a Gabor function (ie. the product of a sinusoid and a Gaussian), unless alpha=0, and ||gON|| = ||gOFF||. Third, the unbalanced combination of the on and off fields (when alpha>0) would make the RFs sensitive to luminance, as correctly implied by L.751. So presumably for the contrast-only channel, alpha=0, and not 0.1 as stated in L. 750? Fourth, for the contrast-luminance channel, I imagine that you would want to build-in a greater response to dark than light (given what was said in the Introduction about asymmetry in the response to increments and decrements). But this would surely imply alpha<0 in Eqn 8, not alpha>0 (L 750) ? In short, a good deal of clarification is needed here.

Appendix, L. 755: more precisely, Rg,odd and Rg,even are not "a pair of Gabor filters" but "the responses from a pair of Gabor filters", aren't they ?

**Have all data underlying the figures and results presented in the manuscript been provided?**

Reviewer #1: Yes

Reviewer #2: Yes

Reviewer #3: Yes

PLOS authors have the option to publish the peer review history of their article (what does this mean?). If published, this will include your full peer review and any attached files.

Reviewer #1: No

Reviewer #2: **Yes: **Laurent Perrinet

Reviewer #3: No
---

## [Decision Letter · Decision Letter 1]

4 Feb 2021

Dear Drs. Lerer, Super, and Keil,

Thank you very much for submitting your manuscript "Dynamic Decorrelation as a unifying principle for explaining a broad range of brightness phenomena" for consideration at PLOS Computational Biology. As with all papers reviewed by the journal, your manuscript was reviewed by members of the editorial board and by several independent reviewers. The reviewers appreciated the attention to an important topic. Based on the reviews, we are likely to accept this manuscript for publication, providing that you modify the manuscript according to the review recommendations.

Sincerely,

Michael Rudd

Guest Editor

PLOS Computational Biology

Lyle Graham

Deputy Editor

PLOS Computational Biology

Addiional comments from the Guest Editor:

Dear Authors:

The reviewers and I all agree that you have done an excellent job of revising your manuscript and that the revised manuscript is clearly worthy of publication in PLOS Computational Biology.

The only substantive change that has been suggested by any of us is from Reviewer 2, who suggests that the global explanation of the array of lightness illusions that you address might be further clarified. I agree that the explanation might be made pithier, but you do address the mechanisms responsible for the success of you model in the current version of your ms. So I leave it up to you whether you want to make such a change. In case you might, I am classifying your manuscript as "Accepted with Minor Revisions." If you decide to make changes other than fixing minor typos, you will need to submit another revised manuscript with all changes highlighted so that I can approve the changes. If the changes are complex, I might also have Reviewer 2 take a look at them. But I think the paper is excellent as it is and could be accepted in it's current form.

Congratulations and best regards,

Michael Rudd

Minor matters (indexed by line #):

197 “ifor” should be “for”

After all equations no indentation unless it’s a new paragraph

Fig 9 caption: typo: “continuous”

Fig 17 title: missing hyphen in “Craik-O’Brien-Cornsweet”

546 Suggestion: strike “We also wish to note that” before “We were unable…”

Fig 19 caption: No caps for “pyramid”

559 strike comma

607 “gets” should be “becomes”

700 “By all the cheers” sounds odd in English. Maybe “Despite all of these successes?’

782 “1 : 2” should be “1:2”

Reviewer's Responses to Questions

**Comments to the Authors:**

Reviewer #1: The authors have done a very thorough and competent job in this revision. Happy to see it published.

Reviewer #2: In this revision of their paper, authors propose changes in the form of writing but also in the scope of the results that are presented. As such, the manuscript is greatly improved from the first revision and requires only minor revisions before publication. I would like to congratulate the authors once again for this excellent work.

First of all, the response to the various reviewers is detailed and precise. The decisions that have been taken, notably concerning the choice of title or the formalization of concepts in the paper (such as the use of gain normalization) have greatly contributed to improving the quality and clarity of the paper. Notably, both in the answer and in the manuscript, I appreciated the effort that was put into explaining with the model as many luminance illusions as possible while keeping the number of parameters in the model as low as possible.

One point that could be improved is the global explanation of the phenomena leading to luminance illusions throughout the different examples. For example for the Benary cross, the illusion is due to the redundancy of the contours and its modulation by the decorrelation mechanism. Insisting on the overall mechanism that produces the illusion will amplify the impact of the manuscript on its audience by explaining the mechanisms underlying the derivation of this particular model.

minor :

138 : "Barkan, [54], " > "Barkan [54] " l165 "White, [66], " > "White [66] " (see reviewers comment in previous revision)

223: g should be in math mode, not between apostrophes

225 : " image ." > " image."

774 : ", " at the begin of the line

Reviewer #3: Overview

The authors have made extensive revisions to the MS that have improved its coherence and clarity. I think that their novel synthesis of spatial filtering, contrast gain control and image reconstruction ('filling-in') to account for brightness perception is plausible, interesting and important, and could have quite an impact on the field. I'm broadly satisfied with the paper as it now is, though a few corrections still need doing, as below. There may of course be others as well.

Corrections:

Legend to Figs. 7A, 9C, 10C, 11C, 13A, 14A, 16A, 17A, 19C, 20B, 22B need correction (update) to the stated line colours: black, blue >>> red, black (arising from changes that I suggested previously for the figure plotting)

Fig 16A 'Duengeon' >>> 'Dungeon'

**Have all data underlying the figures and results presented in the manuscript been provided?**

Reviewer #1: Yes

Reviewer #2: Yes

Reviewer #3: Yes

PLOS authors have the option to publish the peer review history of their article (what does this mean?). If published, this will include your full peer review and any attached files.

Reviewer #1: No

Reviewer #2: **Yes: **Laurent U Perrinet

Reviewer #3: No
---

## [Editor Report · Decision Letter 2]

6 Apr 2021

Dear Leher, Super, and Keil,

We are pleased to inform you that your manuscript 'Dynamic Decorrelation as a unifying principle for explaining a broad range of brightness phenomena' has been provisionally accepted for publication in PLOS Computational Biology.

Best regards,

Michael Rudd

Guest Editor

PLOS Computational Biology

Lyle Graham

Deputy Editor

PLOS Computational Biology

Nice job! It's a nice paper that should have a significant impact. You did an excellent job of revising it and the reviewers were also very helpful. I think the main points of the paper are now significantly clarified, especially relative to the initial submission.

Best,

Michael Rudd

---

## [Editor Report · Acceptance letter]

21 Apr 2021

PCOMPBIOL-D-20-00634R2 

Dynamic Decorrelation as a unifying principle for explaining a broad range of brightness phenomena

Dear Dr Lerer,

I am pleased to inform you that your manuscript has been formally accepted for publication in PLOS Computational Biology. Your manuscript is now with our production department and you will be notified of the publication date in due course.

With kind regards,

Katalin Szabo
